

# Geodetic point surface mass balances: A new approach to determine point surface mass balances from remote sensing measurements

*Christian Vincent[1], Diego Cusicanqui[1,4], Bruno Jourdain[1], Olivier Laarman[1], Delphine Six[1], Adrien Gilbert[1], Andrea Walpersdorf[2], Antoine Rabatel[1], Luc Piard[1], Florent Gimbert[1], Olivier Gagliardini[1], Vincent Peyaud[1], Laurent Arnaud[1], Emmanuel Thibert[3], Fanny Brun[1] and Ugo Nanni[1]*

[1] Université Grenoble Alpes, CNRS, IRD, Grenoble-INP, Institut des Géosciences de l'Environnement (IGE, UMR 5001), F-38000 Grenoble, France.

[2] Université Grenoble Alpes, CNRS, ISTerre, Grenoble, France.

[3] Université Grenoble Alpes, INRAE, UR ETGR, Grenoble

[4] Université Savoie Mont-Blanc, CNRS, Laboratoire EDyTEM, F-73000 Chambery, France

Corresponding author: Christian Vincent (christian.vincent@univ-grenoble-alpes.fr)

**Abstract**

Mass balance observations are very useful to assess climate change in different regions of the world. As opposed to glacier-wide mass balances, which are influenced by the dynamic response of each glacier, point mass-balances provide a direct climatic signal that depends on surface accumulation and ablation only. Unfortunately, major efforts are required to conduct *in situ* measurements on glaciers. Here, we propose a new approach that determines point surface mass balances from remote sensing observations. We call this balance the geodetic point surface mass balance. From observations and modelling performed on Argentière and Mer de Glace glaciers over the last decade, we show that the vertical ice flow velocity changes are small in areas of low bedrock slope. Therefore, assuming constant vertical velocities in time for such areas and provided that the vertical velocities have been measured for at least one year in the past, our method can be used to reconstruct annual point surface mass balances from surface elevations and horizontal velocities alone. We demonstrate that the annual point surface mass balances can be reconstructed with an accuracy of about 0.3 m w.e. a$^{-1}$ using the vertical velocities observed over the previous years and data from Unmanned Aerial Vehicle images. Given the recent improvements of satellite sensors, it should be possible to apply this method to high spatial resolution satellite images as well.



## 1. Introduction

Glacier surface mass balance observations are widely used to assess climate change in various climatic regimes because of their sensitivity to climate variables [e.g. Zemp et al., 2019; Marzeion et al., 2014; Kaser et al., 2006; Gardner et al., 2013; Huss and Hock, 2018; IPCC, 2019 ]. *In situ* surface mass balance measurements have been conducted on only a few of the 200,000 mountain glaciers worldwide [WGMS, 2017; Zemp et al., 2015]. In the European Alps, about a dozen annual surface mass balance time series from *in situ* measurements extending over more than 50 years are available [WGMS, 2017]. Recently, considerable efforts have been made to assess ice volume changes at the mountain-range scale over long time periods using geodetic measurements obtained from remote sensing techniques [e.g. Paul and Haeberli, 2008; Abermann et al., 2011; Gardelle et al., 2012; Gardner et al., 2013; Berthier et al., 2014; Brun et al., 2017]. These geodetic methods determine glacier-wide volume changes, or glacier-wide mass balances, by differencing repeated determinations of glacier surface elevations obtained from airborne and spaceborne surveys, usually over multiyear to decadal periods [e.g. Vincent, 2002; Bauder et al., 2007; Soruco et al., 2009; Berthier et al., 2014; Dussaillant et al., 2019]. These methods are effective to estimate the overall glacier mass change and quantify the related hydrological impacts or sea level contribution [e.g. Hock et al., 2005; Kaser et al., 2010; Huss, 2011; Immerzeel et al., 2013; Zemp et al., 2019]. However, the meaningfulness of a climatic interpretation of these results is questionable. Indeed, glacier-wide mass balances are not solely driven by changes in climate but also by changes in glacier geometry controlled by the dynamic response of each glacier [Vincent, 2002; Fischer et al ., 2010; Abermann et al., 2011; Huss et al., 2012; Vincent et al., 2017 ]. Consequently, they do not provide a direct climatic signal. On the other hand, point surface mass balances provide a direct climatic signal which depends only on local accumulation and ablation (Huss and Bauder, 2009; Thibert et al., 2013; Vincent et al., 2004, 2017, 2018b). However, the only way to presently obtain point mass balance data is to make *in situ* measurements. In particular, the net annual ablation in the ablation zone is usually obtained from ablation stakes. These point surface mass balance measurements require huge efforts involving field campaigns and the collection of data from stake measurements scattered over the glacier. This explains why so few *in situ* measurements are performed, especially on glaciers located in remote areas with very difficult access (e.g., Azam et al., 2018; Wagnon et al., 2013; Hoezle et al., 2017).

The objective of this paper is to propose an approach to determine point surface mass balances from measurements obtained by remote sensing techniques. Our aim is to determine point surface mass balances in ablation areas without setting up ablation stakes each year. We will develop this method using a comprehensive dataset of *in situ* measurements and analysis of ice motion, elevation changes and point surface mass balance data in the ablation area of Argentière Glacier (French Alps). We will then validate our method in other areas of the ablation zone of this glacier and of the Mer de Glace glacier.

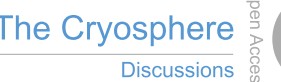

## 2. Study area

The Argentière Glacier is located in the Mont-Blanc range, French Alps (45°55' N, 6°57'E). Its surface area was about 12.4 km² in 2003 (Fig. 1). The glacier extends from an altitude of about 3,400 m a.s.l. at the upper bergschrund down to 1,600 m a.s.l. at the snout. The length of this glacier is about 10 km. It faces north-west, except for a large part of the accumulation area (south-facing tributaries). The annual surface mass balance ranges roughly from 2 meters of water equivalent per year (m w.e. a$^{-1}$) in the accumulation area to about –10 m w.e. a$^{-1}$ close to the snout. This glacier is free of rock debris except for the lowermost part of the tongue, below the ice fall located between 2,000 and 2,300 m a.s.l. The field observations of the Argentière Glacier (i.e. mass-balance, thickness variations, ice-flow velocities and length fluctuations over 50 years) come from the French glacier monitoring program called GLACIOCLIM (Les GLACIers, un Observatoire du CLIMat; https://glacioclim.osug.fr/). For the present study, additional detailed observations were carried out in the framework of the SAUSSURE program (Sliding of glAciers and sUbglacial water pressure (https://saussure.osug.fr). The main part of our study focuses on a small area of Argentière Glacier (∼0.2 km²) located at 2,350 m a.s.l. in the ablation zone (Fig. 1 and 2). In this area, the glacier is ∼600 m wide, the horizontal ice flow velocity is ∼55 m a$^{-1}$ (Vincent and others, 2009) and the maximum ice thickness is 250 m (Rabatel et al., 2018). Experiments conducted in boreholes (Hantz and Lliboutry, 1983) indicate that the bed is composed of hard rock with no thick and deforming sediment layer.

## 3. Data

In the selected area, point annual surface mass balances and ice flow velocities were monitored accurately at the end of each ablation season between 2016 and 2019 from 19 ablation stakes (Fig. 2). Our study also used surface mass balance and ice flow velocity observations from a small part of the ablation zone of the Mer de Glace glacier, at the location named "Tacul glacier" (Fig.1).

The ablation stakes are 10 meters long and made of five 2-m long sticks tied together with metallic chains. Errors in ablation measurements mainly come from the mechanical play of the jointed sticks. The uncertainties of the annual surface mass balance measurements performed in this ablation zone have been assessed at 0.14 m w.e. a$^{-1}$ (Thibert et al., 2008). Topographic measurements were performed to obtain the 3D coordinates of the ablation stakes. For this purpose, we used a Leica 1200 Differential Global Positioning System (GNSS) receiver, running with dual frequencies. Occupation times were typically one minute with 1-second sampling and the number of visible satellites (GPS and GLONASS) was greater than 7. The distance between fixed and mobile receivers was less than 1 km. The DGPS positions have an intrinsic accuracy of ± 0.01 m. However, given the size of the holes drilled to insert the stakes, we estimate that the stake positions have an uncertainty of ±0.05 m.



The vertical velocity is the vertical component of the surface velocity obtained from stake
measurements. It is obtained from the altitude differences of the bottom tip of the stake. In practice, the
DGPS measurements are performed simultaneously with the emergence measurements in order to obtain
the exact position of the bottom tip of the stake buried in ice. In this way, it is possible to monitor ice
velocity along the three coordinate directions. Depending on the tilt of the ablation stakes, the size of
the drilling hole and the mechanical play of the jointed stakes, we assume that the annual horizontal and
vertical velocities are known with an uncertainty of $\pm$ 0.10 m a$^{-1}$.
Aerial photographs of the glacier surface were taken on 5 September 2018 and 13 September 2019 using
the senseFly eBee+ Unmanned Aerial Vehicle (UAV). A total of 720 photos in 2018 and 673 photos in
2019 were collected with the onboard senseFly S.O.D.A. camera (20 Mpx RGB sensor with a 28 mm
focal lens from an average altitude of 140 m above the glacier surface). Prior to the survey flights, we
collected GNSS measurements of ground control points (GCPs) that consist in rectangular pieces of red
fabric (100x60 cm) with white painted circles (40 cm diameter) on the glacier (10 in 2018, 20 in 2019)
and ten 40 cm diameter white circles painted on rocks on the sides of the glacier. The original horizontal
resolutions of the ortho-photo mosaics and digital elevation models (DEMs) are 10 cm and 1.00 m,
respectively. The photos from the survey were processed using the Structure for Motion (SfM) algorithm
that is implemented in the Agisoft Metashape Professional version 1.5.2 software package (Agisoft,
2019). The SfM stereo technique was then used to generate a dense point cloud of the glacier surface.
This dense point cloud was used to construct the DEMs using the GCPs surveyed during the field
campaigns. A detailed description of the processing steps can be found in Kraaijenbrink et al. (2016) or
Brun et al. (2016).
To calculate horizontal ice flow velocities over the studied area, we used the UAV ortho-photo mosaics
with COSI-Corr (Co-registration of Optically Sensed Images and Correlation), a software tool
developed for image correlation (Leprince et al., 2007; Ayoub et al., 2009). Due to the velocities of the
Argentière glacier in this region (~55 m a$^{-1}$), we resampled the UAV ortho-photo at 0.1 m resolution
because the correlation was too noisy even with very large window sizes (i.e. 512 pixels). The surface
velocities were computed using an initial window size of 256 pixels, a final window size of 64 pixels
and a step of 4 pixels. The output velocity field was filtered using signal-to-noise ratios (SNR) provided
by COSI-Corr. Using an SNR threshold provides a good compromise between output details, noise and
computing time. A detailed description of the correct choice of the window size for correlation can be
found in Kraaijenbrink et al. (2016).
To establish the possible errors on the correlation process, horizontal displacements on stable off-glacier
areas were evaluated and provided a maximum horizontal error of ~0.5 m.



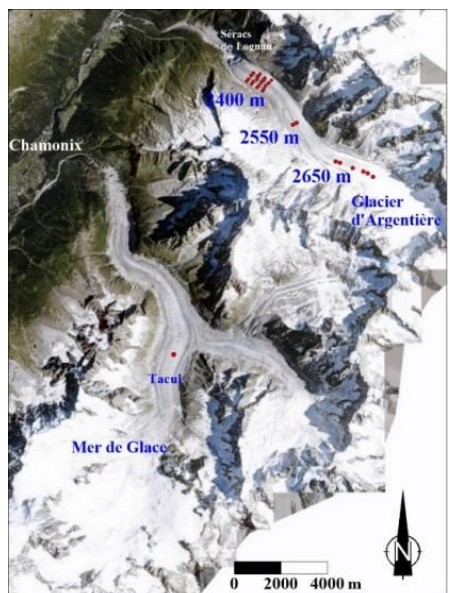


*Figure 1: Map of Argentière and Mer de Glace glaciers. The red dots on Argentière glacier are the*
*ablation stakes used in this study for annual surface mass balance and ice flow velocity measurements*
*in 3 regions of the glacier (at approximately 2,400; 2,550 and 2,700 m a.s.l.). Aerial photo from the*
*French National Geographical Institute, 2015 (https://www.geoportail.gouv.fr/).*

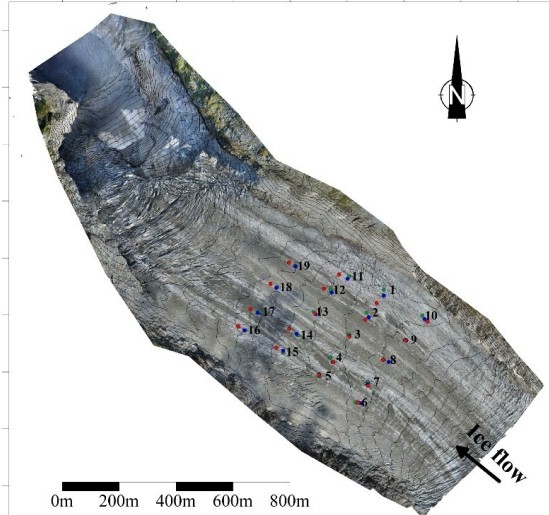


*Figure 2: Map of the studied area in the ablation zone of Argentière glacier. The contour lines of surface*
*topography correspond to the surface in 2018. The green, blue and red dots are the positions of the*
*ablation stakes used for surface mass balance and ice flow velocity measurements when they were set*
*up in 2016, 2017 and 2018, respectively. Aerial photo from Unmanned Aerial Vehicle survey (5*
*September 2018).*







## 4.   Method

We will now introduce the mathematical frameworkused further on.

### 4.1 Emergence velocities

The emergence velocity is the upward or downward flow of ice relative to the glacier surface. This flow
compensates the surface mass balance exactly if the glacier is under steady state conditions. The surface
elevation change equation (Cuffey and Paterson, 2010, p. 332) expresses the surface mass balance as a
function of surface velocity and surface gradient:

$b_s = \partial S / \partial t - w_s + u_s\, \partial S / \partial x + v_s\, \partial S / \partial y$             (1)

with $b_s$ the surface mass balance expressed in meters of ice, firn or snow (m a$^{-1}$), $S$ the surface elevation
(m), $u_s,\ v_s,\ w_s$ the components of ice flow velocity at the surface (m a$^{-1}$), $\partial S / \partial x$ the surface gradient in the
x direction and $\partial S / \partial y$ in the y direction.
The term $w_s - u_s\, \partial S / \partial x - v_s\, \partial S / \partial y$ is called the emergence velocity. If the horizontal x-axis is taken in the
downslope direction, $v_s = 0$, and the emergence velocity is written as:
$v_e = w_s - u_s\, \partial S / \partial x$                    (2)

Note that, under steady state conditions, $\partial S / \partial t = 0$ and $b_s = -v_e$. The emergence velocities can be
calculated for each ablation stake from horizontal and vertical velocities and the slope of the surface
$\partial S / \partial x$. The slope of the surface can be obtained from GNSS field measurements and calculated over a
distance similar to that travelled by the stake over one year. In the ablation zone, the emergence
velocities are positive, which corresponds to an upward flow of ice relative to the glacier surface. Note
also that the vertical velocity can be positive or negative on any region of the glacier. The emergence
velocity is a classical way to relate the surface mass balance to the thickness changes (Eq. 1).
Unfortunately, as shown later in our study, even if the horizontal and vertical velocities are known
accurately, the large uncertainties related to the slope and thickness changes prevent us from calculating
the point surface mass balance from the emergence velocities.
At the scale of the year, according to Equation 1 and Figure 3, and considering that the x-axis is taken
along the flow line direction (*i.e.*, $v_s = 0$), the annual surface mass balance $B_s$ between the years t and
t+1 is obtained from:



$B_s = \Delta h_1 + U_s \tan \alpha_{t+1} - W_s = \Delta h_2 + U_s \tan \alpha_t - W_s$        (3)

where $u_s.\partial S/\partial x$ is replaced by $U_s \tan\alpha_t$ or $U_s. \tan \alpha_{t+1}$ and $U_s$ is the annual surface horizontal velocity and
$\tan\alpha_t$ and $\tan \alpha_{t+1}$ are the slopes for the years t and t+1 respectively. $W_s$ is the annual vertical velocity.
$\partial S/\partial t$ is replaced by $\Delta h_1$ and $\Delta h_2$, which are the annual thickness changes observed at the ends of the
annual ice flow vector.
Figure 3 illustrates the components of Equation 3.
Note that the slope of the surface may change from year t to year t+1 and the expression depends on the
selected slope and thickness changes $\Delta h_1$ or $\Delta h_2$ (Fig. 3). Obviously, the results are the same.


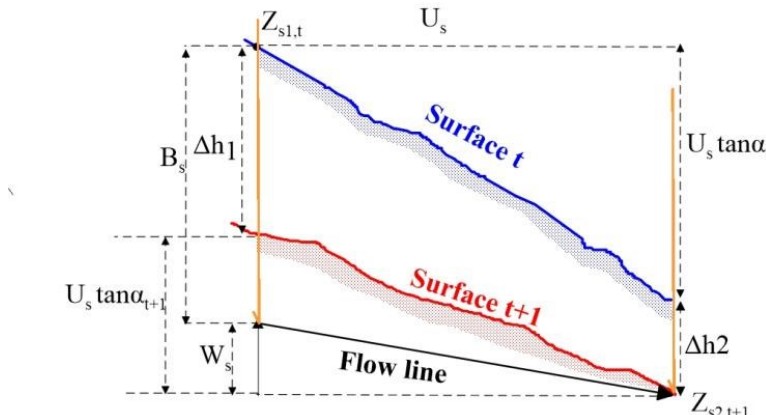


*Figure 3: Diagram illustrating horizontal, vertical and emergence velocities observed from an ablation*
*stake (orange). $U_s$ and $W_s$ are the components of horizontal and vertical velocities, $\alpha_t$ and $\alpha_{t+1}$ the slopes*
*for the years t and t+1 respectively, $Z_{s1,t}$ and $Z_{s2,t+1}$ the elevations of the surface at each end of the ice*
*flow vector and $\Delta h_1$ and $\Delta h_2$ the elevation changes at each end of the ice flow vector.*

208        **4.2 Calculation of the "geodetic point surface mass balance"**


Let us reconsider the emergence velocity formulation in order to express the point surface mass balance
as a function of vertical velocity and altitude changes at the ends of the annual displacement vector.
According to Equation 3 and given that $\Delta h_1 + U_s \tan \alpha_{t+1} = \Delta h_2 + U_s \tan \alpha_t = Z_{s2,t+1} - Z_{s1,t}$ (Fig. 3), we
can write:

$B_s = Z_{s2,t+1} - Z_{s1,t} - W_s$            (4)



This expression has a great advantage in that it does not depend on the surface slope that can change
from one year to the next. It is also independent of thickness changes that can change from one site to
another.
The term geodetic point surface mass balance refers to the value of $B_s$ obtained from Equation 4. Once
the vertical velocity is known, $B_s$ can be obtained from topographical surface measurements alone. Note
that even if the horizontal velocity is not included in Equation 4, it is needed to estimate the positions at
which $Z_{s2,t+1}$ and $Z_{s1,t}$ should be measured.

**5.  Results**

**5.1 Annual horizontal and vertical velocities over the three years**

Annual horizontal and vertical velocities were measured from a network of 19 ablation stakes over three
years between 2016 and 2019 (Fig. 2). The stakes were replaced each year and were always set up at the
same locations, using a handheld GPS device, allowing a relevant comparison, except for stakes 1 and
11 which were located in areas with large crevasses, preventing the possibility of drilling stakes at the
chosen location. In addition, for the year 2018/2019, stake 12 was accidentally replaced at a distance of
more than 30 meters from the initial position due both to a lack of rigour and to the uncertainty of the
handheld GPS instrument. This error led to a bias in the horizontal velocity of 3 m a$^{-1}$ in a region with a
strong horizontal gradient (left edge of the area in Figure 4). However, it does not change the pattern of
horizontal velocities or horizontal velocity changes with time. This is not the case for vertical velocities
as shown below. In the area of this network, the annual horizontal velocities range from 35 to 60 m a$^{-1}$.
The annual ice flow velocities have been interpolated from kriging over the entire coloured areas shown
in Figure 4. In this way, we can accurately compare the ice flow velocities over three years, 2016/2017,
2017/2018 and 2018/2019, at the locations of each stake (Fig. 5a). Strong deceleration in horizontal ice
flow velocities can be observed over these three years. On the average, ice flow velocity decreased by
2.4 and 1.8 m a$^{-1}$ over the two periods, which corresponds to an average decrease of about 4.8 and 3.6%
per year, respectively. Note that the regression lines shown in Figure 5a are almost parallel, which means
that the change in velocities is homogeneous in space.
The vertical velocities were obtained from the altitude changes of the bottom tip of the stakes from one
year to the next (Fig. 3). In the studied area, the vertical velocities can be positive or negative and range
from -4 to 4 m a$^{-1}$ (Fig. 4). The vertical velocities have been interpolated over the entire coloured areas
shown in Figure 4 using kriging. The patterns of vertical velocities are very similar for the year
2016/2017 and 2017/2018. We note some differences with the 2018/2019 pattern. As mentioned
previously, stakes 1, 11 and 12 set up in 2018/2019 are located at distances of more than 30 meters from
the initial positions. In addition, stakes 17, 18, 19 were replaced in 2018 at distances ranging between
25 and 30 meters from the initial positions. These six stakes are shown with small dots in Figure 5b. If





we exclude the velocity values of 2018/2019 for these stakes, we can conclude that the measured vertical
velocities are very similar over this 3-year period. The differences do not exceed 0.5 m a$^{-1}$. The average
of the differences is 0.01 m a$^{-1}$ and the standard deviation is 0.29 m a$^{-1}$. These differences barely exceed
the measurement uncertainty. Note also that the vertical velocity changes could be affected by the
horizontal motion changes or vertical strain rate changes as discussed in Section 6.






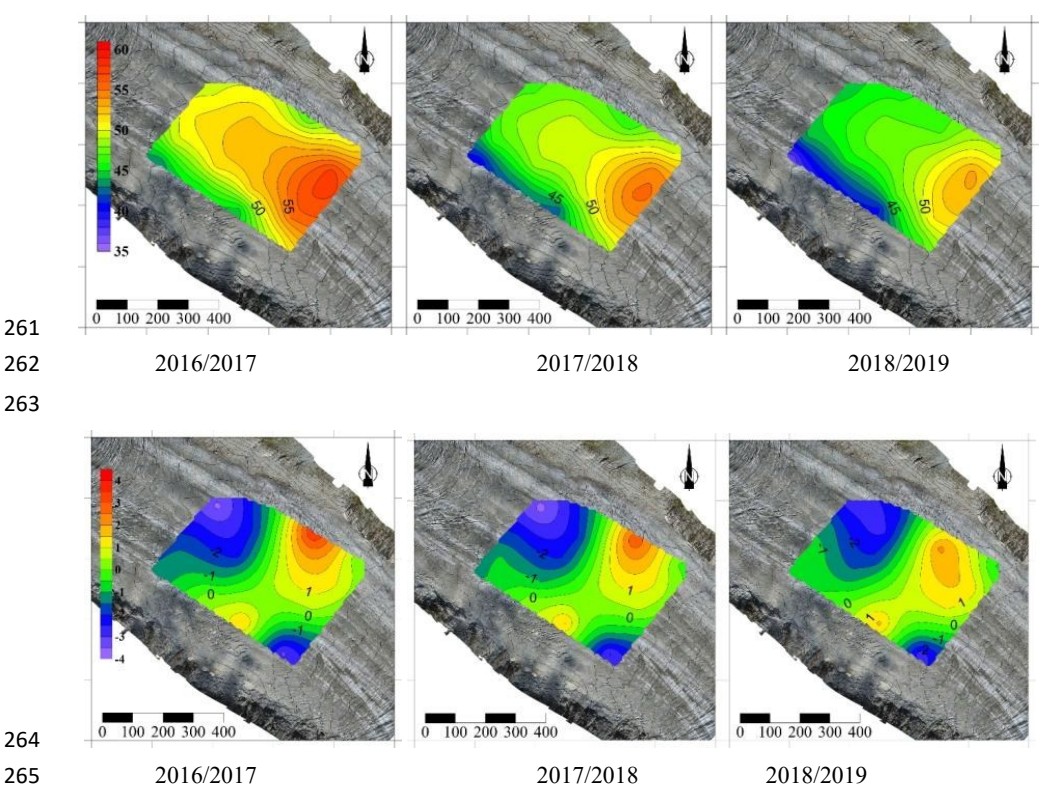

Figure 4: Horizontal (top panel) and vertical (bottom) ice flow velocities (m a$^{-1}$) measured over three years from the ablation stakes. Note the different colour scales. Distances in m.

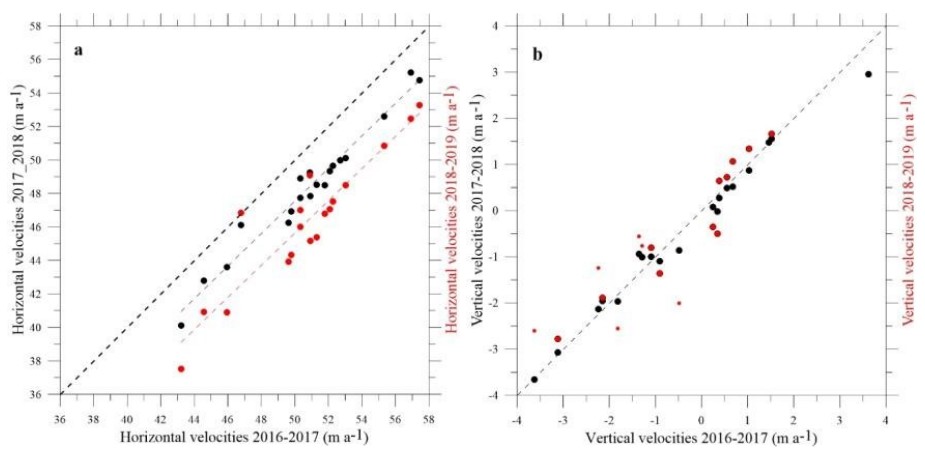

Figure 5: Comparison of horizontal ice flow velocities (a) and vertical velocities (b) between the years 2016/2017, 2017/2018 and 2018/2019. The black dots correspond to the comparison between the



*2016/2017and 2017/2018 periods. The red dots correspond to the comparison between the 2016/2017*
*and 2018/2019 periods. The thick dashed line corresponds to the bisector and the thin dashed lines to*
*the regression lines. The small dots in the figure on the right  correspond to the stakes that were set up*
*in 2018 at distances of more than 25 m from the initial positions.*


**5.2 Emergence velocities**

The emergence velocities have been calculated from Equation 2 for each stake and reported in Figure 6.
We compared the emergence velocities obtained each year at each stake location (Fig. 7). Unlike the
vertical velocities, the differences between emergence velocities calculated over the 3 years reveal a
standard deviation of 0.8 m w.e. a$^{-1}$. The value of emergence velocities is affected by large uncertainties
related to the slope.
Combined with the measured thickness changes, the emergence velocity should make it possible to
estimate the surface mass balance. However, our study shows that the uncertainties in the emergence
velocity prevent us from calculating the point surface mass balance accurately. Indeed, the dispersion
of 0.8 m w.e. a$^{-1}$ is large compared to the spatial variability of about 1 m w.e. a$^{-1}$ for point surface mass
balance in the ablation zone of alpine glaciers (Vincent et al., 2018b).
For this reason, to calculate the surface mass balance, we suggest using the "geodetic point surface mass
balance" described earlier rather than the emergence velocity.

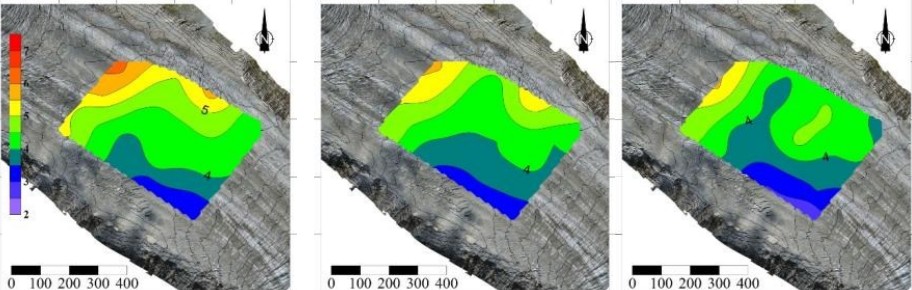


*Figure 6: Emergence velocities between the years 2016/2017, 2017/2018 and 2018/2019 (m a$^{-1}$)*


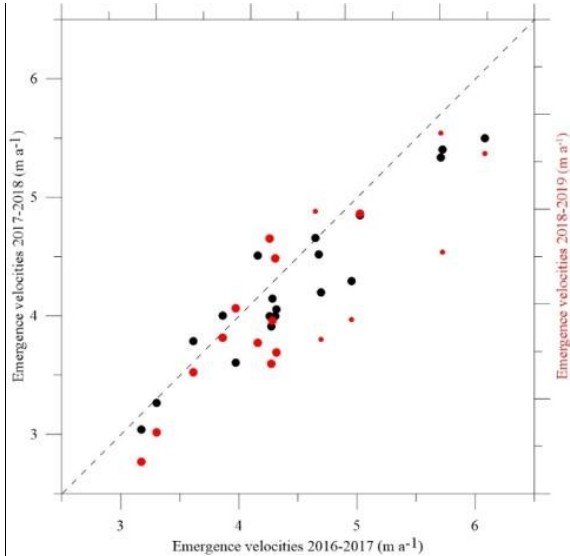


*Figure 7: Comparison of emergence velocities between the years 2016-2017, 2017-2018 and 2018-2019. The black dots correspond to the comparison between the 2016-2017and 2017-2018 periods. The red dots correspond to the comparison between the 2016-2017 and 2018-2019 periods. The red small dots correspond to the stakes that were set up in 2018 at distances of more than 25 m from the initial positions*

305
306
307

### 5.3 "Geodetic point surface mass balances" using *in situ* GNSS measurements

The geodetic point surface mass balance is calculated according to Equation 4. We first tested the method in the studied region of Argentière glacier at 2,400 m a.s.l. using the *in situ* GNSS measurements. For this purpose, we used the altitudes of the surface at the stake locations for the years 2017 and 2018 and the vertical velocities observed in 2016-2017. The resulting point surface mass balances for the hydrological year 2017-2018 are compared with the observed surface mass balance and plotted in Figure 8a. Note that the surface mass balances are in m of ice per year. The comparison shows very good agreement. The maximum difference is 0.39 m of ice per year and the standard deviation is 0.20 m of ice or 0.18 m w.e. per year. In addition, we calculated the surface mass balances of 2018-2019 from the vertical velocities observed in 2016-2017 and 2017-2018 (Fig. 8b). In this case, the comparison with the observed surface mass balances shows large discrepancies. However, a more detailed analysis reveals that the calculated and observed surface mass balances are very similar if the vertical velocities observed in 2016-2017 and 2017-2018 were measured exactly at the same location of the stakes measured in





2018-2019. In Figure 8b, the large dots show the calculated and observed surface mass balances for the
stakes located within a distance no greater than 15 m. From this comparison, the differences are less
than 0.5 m a$^{-1}$ of ice and the standard deviation is 0.17 m of ice or 0.15 m w.e. a$^{-1}$.
From this analysis, we conclude that the geodetic point surface mass balance can be obtained with an
accuracy of about 0.2 m w.e. a$^{-1}$ using the vertical velocities observed over the previous years. It requires
measurement of the horizontal ice flow velocity and the altitudes of the ends of the velocity vector
exactly at the same location, within a radius of less than 15 m compared to that of vertical velocity
determination. In practice, the vertical velocities should be observed accurately between two years t and
t+1 from stakes and GNSS measurements. Then, for the following or previous years, the point surface
mass balance can be obtained from surface measurements only (without drillings and setting new stakes)
using the horizontal velocity and the altitudes of the surface measured at each end of the horizontal
vector. In the next section, we examine how such measurements obtained from remote sensing data can
also be used effectively to determine the point surface mass balance.


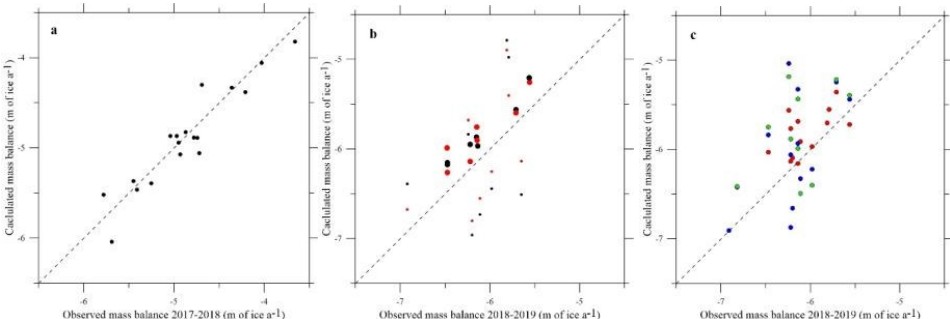



*Figure 8: Observed and calculated point surface mass balances at 2,350 m a.s.l. at Argentière glacier.*
*The point surface mass balances have been calculated: a) for the year 2017-2018 using the vertical*
*velocities measured in 2016-2017 and elevations from GNSS measurements; b) for the year 2018-2019*
*using the vertical velocities measured in 2016-2017 (black dots) and 2017-2018 (red dots), and*
*elevations from GNSS measurements; c) for the year 2018-2019 using elevations from remote sensing*
*data (UAV data) and the vertical velocities measured in 2018-2019 (red dots), 2017-2018 (blue dots)*
*and 2016-2017 (green dots). The large dots shown in Figure 8b correspond to the stakes which were*
*set up within a radius of less than 15 m.*









### 5.4 "Geodetic point surface mass balances" using remote sensing measurements

Here, we used the same method described in the previous section. However, the *in situ* GNSS measurements used to determine the altitudes and horizontal velocities are replaced by remote sensing measurements. For this purpose, we used the horizontal velocities (Figure 9) and the DEMs (Figure 10) obtained from UAV surveys in 2018 and 2019. The vertical velocities are those observed in 2018-2019, 2017-2018 and 2016-2017. The horizontal velocities have been neglected for stakes 9 and 10 given the poor quality of the correlation and the opening and/or closing of crevasses in the ice (close to the stake 10) that caused a drastic change between the photos, which subsequently affected the image correlation (Fig. 9).

Some details on the procedure are given below for the sake of clarity. The horizontal velocities retrieved from the UAV surveys were determined at positions where vertical velocities were measured. In this way, the coordinates XY of each vector end have been calculated (green dots on Fig. 9). Then we used the DEMs from 2018 and 2019 to determine the elevations of these points $Z_{s1, 2018}$ and $Z_{s2, 2019}$ (see Eq. 4 and Figure 3). The comparison between the *in situ* horizontal velocities and the velocities obtained from the UAV surveys reveals a standard deviation of 0.7 m a$^{-1}$.

The reconstructed point surface mass balances are compared with the observed surface mass balances in Figure 8c. For this reconstruction, we used the vertical velocities observed in 2018-2019 (red dots), 2017-2018 (blue dots) and 2016-2017 (green dots). For the reconstructions using the vertical velocities of 2016-2017 and 2017-2018, we excluded data of sites 1, 11, 12, 17, 18 and 19 for which the stakes were measured at distances of more than 30 m from those of 2018-2019.

The differences between the observations and the reconstructed surface mass balances using the 2018-2019 vertical velocities are less than 0.45 m of ice per year and the standard deviation is 0.24 m of ice or 0.22 m w.e. a$^{-1}$. The differences between the observations and the reconstructed surface mass balances using the 2016-2017 and 2017-2018 vertical velocities show standard deviations of 0.42 and 0.40 m w.e. a$^{-1}$., respectively.

From these results, we conclude that the point surface mass balances can be obtained with an accuracy of about 0.3 m w.e. a$^{-1}$ using remote sensing measurements, assuming that the vertical velocities have been observed accurately over the previous years.



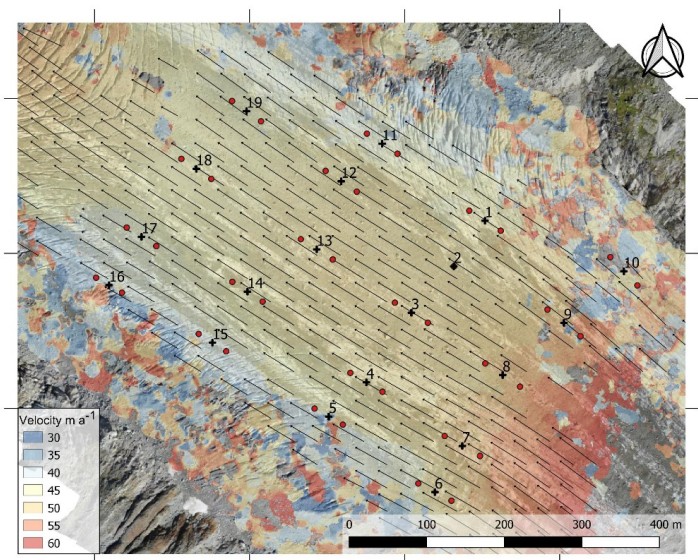

*Figure 9: Horizontal velocities obtained from feature tracking (Cosi-Corr) using UAV images. The black crosses show the locations where the vertical velocities were observed. The red dots correspond to the ends of horizontal vectors for 2018-2019, determined from UAV images.*

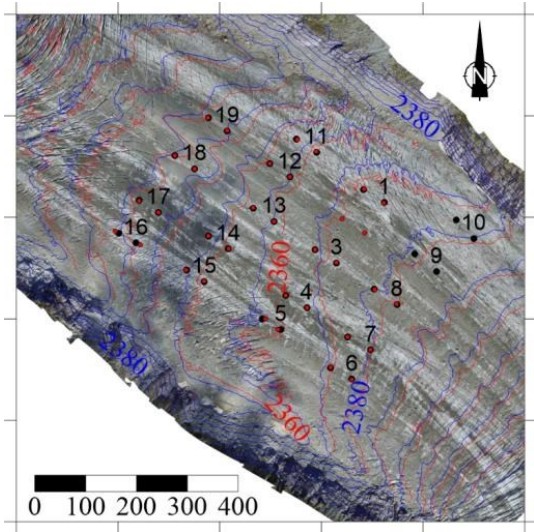

*Figure 10: DEMs obtained from the UAV survey in 2018 (blue contour lines) and 2019 (red contour lines). The black dots correspond to the positions of the stakes in 2018 and 2019 observed from GNSS measurements. The red dots correspond to the ends of the horizontal velocity vectors obtained from UAV images.*





**5.5 Validation of the method: geodetic surface mass balances obtained in other regions**

In order to establish that the results are neither accidental nor site-dependent, we tested the method on
other areas of Argentière glacier and on another glacier: the Mer de Glace, approximately 10 km away
(Fig. 1), for which vertical velocities were available. Here, we used GNSS *in situ* measurements given
that accurate elevations observations from remote sensing data are not available.
First, we selected two ablation stakes in a sector of Argentière glacier located at 2,530 m a.s.l. These
stakes were replaced within a radius of ± 35 m each year between 2001 and 2018 (Fig. 1). Note that
these measurements were not intended for vertical velocity determination but rather for point surface
mass balance measurements. This explains why the stakes were not set up at exactly the same locations
over the whole period. Note also that the region is not debris-covered and consequently the surface
roughness is lower compared to the studied area at 2,350 m. Using Equation 4 and the method described
in the previous section, we calculated the point surface mass balances at these two stakes over the period
2001-2018. For this purpose, we used the average vertical velocities calculated over this period and the
altitudes of each stake for each year of this period. These two stakes (named stake 2 and stake 3) are
located about 120 m apart. The average calculated vertical velocities are -0.24 m a$^{-1}$ (± 0.44 m a$^{-1}$) and
-0.79 m a$^{-1}$ (± 0.33 m a$^{-1}$), respectively, and did not show strong temporal changes (Fig. 11b). Note that
the horizontal velocity decreased from 75 to 50 m a$^{-1}$ in this region between 2002 and 2018 (Fig. 11a).
The geodetic point surface mass balances are compared to the observations (Fig. 12a). The standard
deviations of the calculated and observed surface mass balance differences are similar to those of the
vertical velocities (0.44 and 0.33 m a$^{-1}$, i.e. 0.4 and 0.3 m w.e. a$^{-1}$ ).
Second, we tested the method in another sector of Argentière glacier, close to the equilibrium line, which
is located close to 2,800 m a.s.l. For this purpose, we selected 6 stakes (stakes 7, 8, 9, 10, 11, 12) which
were measured along a longitudinal section between 2,650 and 2,750 m a.s.l. (Fig. 1) over the period
2005-2018. In this region, the horizontal ice flow velocity is about 50 m a$^{-1}$ (Fig. 11a). Here again, the
network of stakes was mainly designed for point surface mass balance measurements. Thus, given that
the stakes were set 10-m deep in the ice and the surface mass balance ranges between -4 and 0 m w.e.
a$^{-1}$ depending on the year, the ablation stakes were not replaced each year. As the ablation stakes move
with the ice flow, we selected only the measurements that were performed at the same locations. Indeed,
after the first year following installation, the location of each stake was far from its initial position and
we cannot assume that the vertical velocity was similar. Consequently, 5 years are available to calculate
the vertical velocities and to make the comparison between calculated and observed point surface mass
balances (Fig. 12b). The standard deviations of calculated and observed point surface mass balance
differences are 0.22 m of ice a$^{-1}$, *i.e.* 0.20 m w.e. a$^{-1}$.



Finally, we tested the method on another glacier, Mer de Glace (Fig. 1). On this glacier, we selected
one stake at 2100 m a.s.l that was measured over 15 years between 2003 and 2018 (Vincent et al.,
2018a). This ablation stake was set up each year at the same location, within a radius of about 30 meters.
Using the method described in the previous sections, we calculated the point surface mass balances at
this stake over the period 2003-2018. The average calculated vertical velocity is -1.10 m a$^{-1}$. Note that
the horizontal velocity decreased from 80 to 50 m a$^{-1}$ and the thickness by 55 m in this region between
2003 and 2018. The results are plotted in Figure 12c. The standard deviation of the calculated and
observed point surface mass balance differences is 0.40 m w.e. a$^{-1}$ .

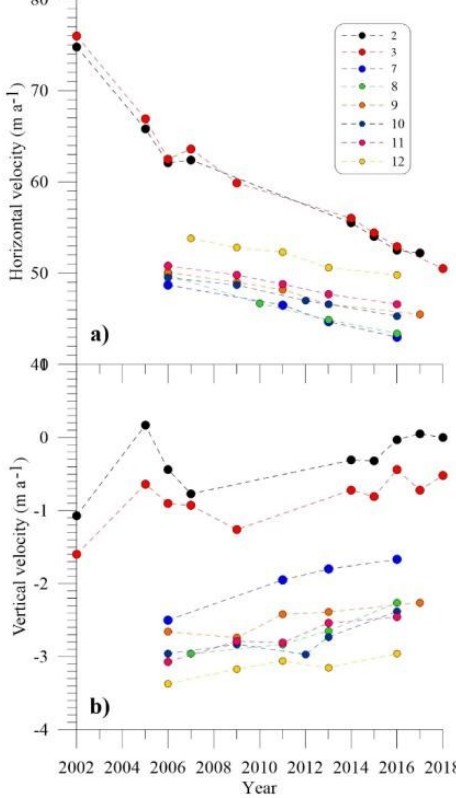

*Figure 11: Horizontal (a) and vertical (b) velocities observed at the different stakes at 2,550 m a.s.l.(Stakes 2 and*
*3) and 2,700 m a.s.l. (stakes 7, 8, 9, 10, 11 and 12).*

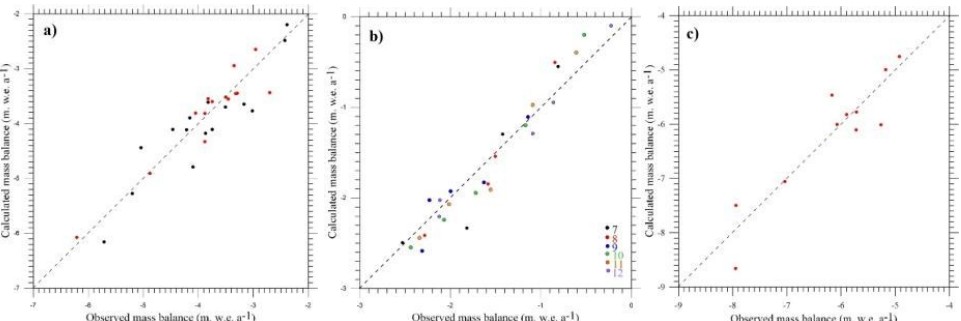

*Figure 12: Observed and calculated point surface mass balances from: a) two ablation stakes located at 2,550 m a.s.l. at Argentière glacier measured between 2002 and 2018, b) six stakes located at around 2,700 m a.s.l. at Argentière glacier measured between 2006 and 2017 and c) one stake located at 2,100 m a.s.l. on Mer de Glace glacier measured between 2003 and 2018.*

## 6. Discussions

### 6.1 Point surface mass balance obtained from emergence velocities *vs.* vertical velocities

A classical approach to relate the point surface mass balance to thickness change is to use the emergence velocity (Cuffey and Paterson, 2010; Kaab and Funk, 1999). From this approach, the point surface mass balance is obtained from the sum of the emergence velocity and the thickness change (Eq. 1). However, the value of the mass balance reconstructed from the emergence velocity depends strongly on the selected surface slope and on thickness change, which both vary considerably with space and time. The value of the slope depends on the choice of the selected distance for the slope calculation and on the roughness of the surface.

In addition, the slope can change significantly from one year to the next. The emergence velocity is therefore not well-defined given that it depends strongly on the spatial and temporal changes of surface roughness, preventing an accurate determination of point surface mass balance as shown in our study.

In contrast, in our analysis, we find that the vertical velocity is almost constant from year to year, at least at a decadal time scale. Thus, we propose to reformulate the emergence velocity formulation (Eq. 1) in order to express the point surface mass balance as a function of vertical velocity and altitude changes at the ends of the annual displacement vector (Eq. 4). In this way, provided that the vertical velocity has been assessed from *in situ* measurements over previous years, the point surface mass balance can be determined from remote sensing measurements alone, outside the period of field measurements. Our results from the detailed studied area at Argentière glacier (2,350 m a.s.l.), for which the observations were designed to accurately determine the vertical velocity, demonstrate that the surface mass balance can be obtained from this method with an accuracy of about 0.2 m w.e. a$^{-1}$ from *in situ* GNSS





measurements and about 0.3 m w.e. a$^{-1}$ using elevations and horizontal velocities obtained from very
high resolution remote sensing data acquired from UAV surveys.

**6.2 Spatial and temporal variability of the vertical velocities**

Our dataset shows that the spatial variability of vertical velocities can be large and strongly varies
depending on the considered area at the glacier surface. For instance, we found strong spatial variability
in the vertical velocity pattern at 2,350 m a.s.l. on Argentière Glacier. Our data suggest that the vertical
velocity spatial gradient can reach 1.5 m a$^{-1}$/100 m in this region. As a consequence, a horizontal
deviation of 10 m could lead to a vertical velocity change exceeding the measurement uncertainty (0.15
m a$^{-1}$). Despite this strong spatial variability, we found small changes in vertical velocities with time.
Below, we discuss the results of a numerical experiment conducted on Argentière Glacier to understand
why.
To analyse the spatial and temporal variabilities of the vertical velocities over the entire glacier, we
performed 3D full-Stokes ice-flow simulations for two different glacier geometries using a surface DEM
measured in 1998 and 2015 and reconstructed bedrock topography (Rabatel et al., 2018). The calculation
is solved using the Elmer/Ice model (Gagliardini et al., 2013). The linear basal friction parameter is
inferred from surface velocity and topography measurements made in 2003 (Berthier et al., 2005) using
the adjoint-based inverse method (Gillet-Chaulet et al., 2012). For each given glacier geometry, we
compute the corresponding flow solution and assume constant friction over time. Therefore, changes in
velocity are only induced by changes in the glacier geometry between 1998 and 2015. We used an
unstructured mesh with a 100 m horizontal resolution, refined down to 10 m in the stake network
monitoring area at 2,400 m a.s.l.
By integrating the mass conservation equation for an incompressible fluid along the vertical axis we can
write:
$$w_s = w_b - \int_{z_b}^{z_s} \frac{\partial u}{\partial x} + \frac{\partial v}{\partial y} \, dz \qquad (5)$$

where $w_b$ is the vertical velocity at the bed, $z_s$ is the surface elevation and $z_b$ the bed elevation. Vertical
velocity at the surface can therefore be viewed as a sum of a component coming from sliding along the
bedrock and a component coming from convergence/divergence of the ice flow integrated over the
glacier thickness. For example, local depression in the bedrock topography creates negative vertical
velocity $w_b$ at the glacier base but also flow convergence that creates positive vertical velocity resulting
in a smoothing of surface vertical velocity $w_s$ by the ice deformation. Figure 13a shows the modelled
vertical surface velocity in 2015. At the scale of the glacier, vertical surface velocities are spatially
heterogeneous due to a combination of bedrock slope and the ice flux divergence/convergence (Fig.
13a). In the model, the basal vertical velocity $w_b$ produced by ice flow along the bedrock can lead to



small scale variability of the basal vertical velocity that can be visible at the surface when sliding velocity
is significant, as modelled around 2,400 m a.s.l. in the studied stake network (Fig. 13a). Bedrock
topography is therefore likely the origin of the observed pattern at 2,400 m a.s.l. (Fig. 4). The pattern
differences between the observations and the modelling results are likely due to bedrock elevation errors.
Although the pattern of horizontal velocities is well reproduced (Fig. 13b), it seems difficult to perfectly
reconstruct the vertical velocities.

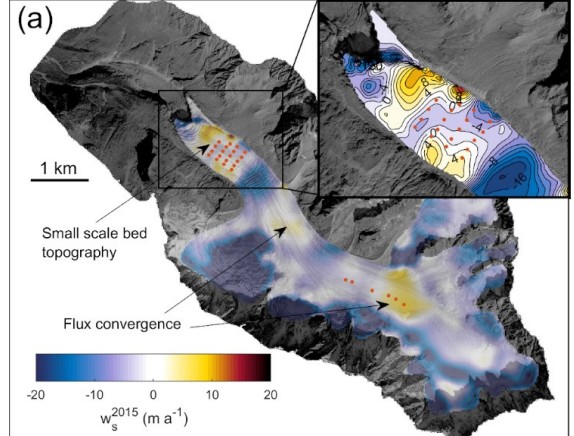
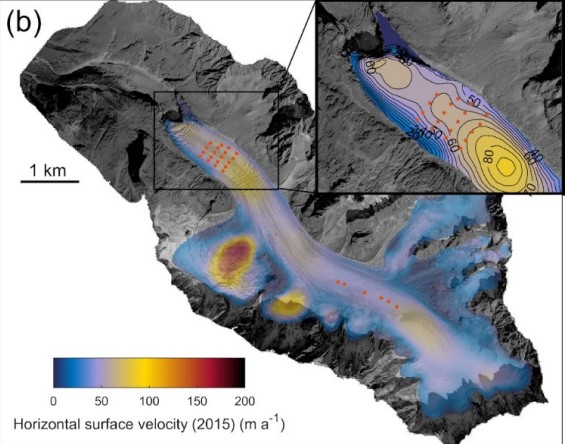


*Figure 13 – Vertical (a) and horizontal (b) surface velocities modelled at Argentière glacier in 2015.*
*Red dots show the locations of the ablation stakes set up at 2,400 m and 2,650 m a.s.l.*

Our numerical experiments were used to analyse the temporal changes in vertical velocities. We found
that the response of the vertical velocities at the glacier surface to changes in glacier thickness over time
is sensitive to the bedrock slope (averaged over a distance greater than the ice thickness). Consequently,
a decreasing vertical velocity magnitude should be associated with decreasing horizontal velocities
where bed slopes are significant (Fig. 14). However, the magnitude of small scale (length-scale inferior
to glacier thickness) spatial variations of vertical velocity due to bedrock topography seems to be little
affected by the large change in horizontal velocities (Fig. 14 and 15). We show that reduced amplitude
of $w_b$ due to decreasing sliding speed is compensated by the reduced amplitude of the ice flux
convergence/divergence produced by bedrock anomalies (red arrows in Fig. 15). Bedrock depressions
and bumps of sizes comparable to glacier thickness produce respectively convergence and divergence
in the ice flow, creating vertical velocities of opposite sign compared to the velocities created by sliding
at the glacier base. These two components of the surface vertical velocity  decrease in magnitude in
response to thickness changes, resulting in a limited change in the sum of the two components and
therefore in surface vertical velocities. This results in nearly constant vertical velocity where large scale



averaged bedrock slope is low, which explains why the observed pattern of surface vertical velocity
(Fig. 4) is well conserved over time.


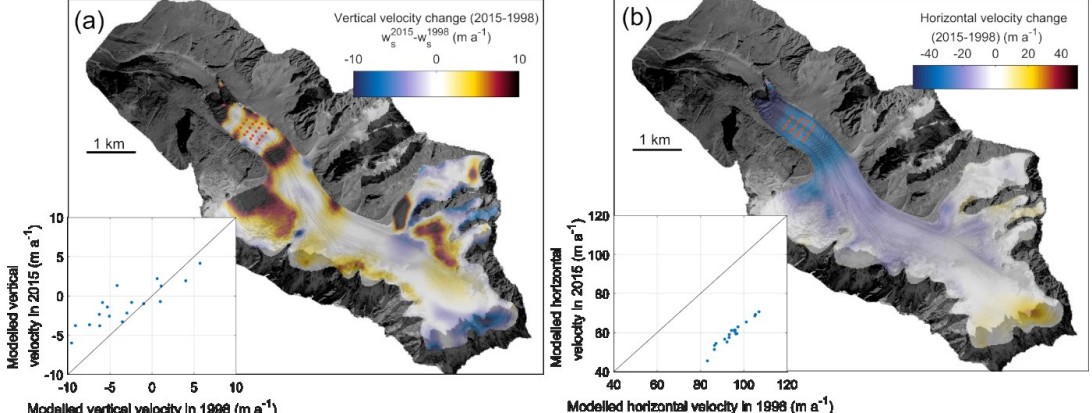


*Figure 14: Modelled changes in vertical (a) and horizontal (b) surface velocities between 1998 and*
*2015. Insets compare modelled velocities at the stake location (orange dots) between 1998 and 2015.*

In summary, at large scale, the magnitude of surface vertical changes over time are proportional to
bedrock slope and changes in horizontal velocities while at small scale, the spatial patterns tend to be
conserved over time due to compensation between changes in bedrock vertical velocities and ice flux
convergence/divergence. These findings suggest that our method is likely applicable only in areas of
low bedrock slope.





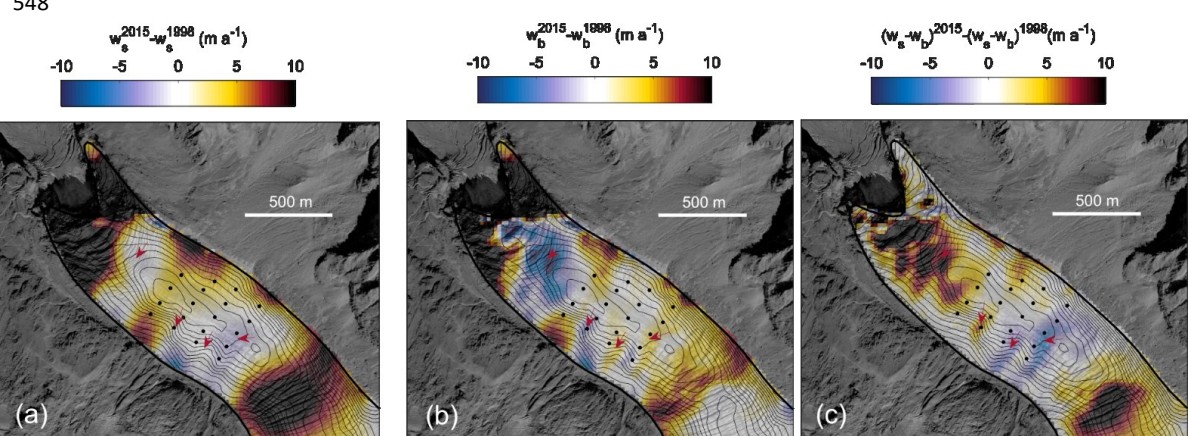

*Figure 15: Modelled changes in vertical velocities at the surface (a) and at the bedrock (b) between*
*1998 and 2015. The right hand figure (c) shows the change in vertical velocity at the surface due to*
*change in flow convergence/divergence. The red arrows indicate the locations where changes in basal*
*vertical velocities are compensated by flow convergence/divergence changes, resulting in constant*
*surface vertical velocities.*

Note that, in our study, we used the annual velocities from September to September. Many studies
pointed out seasonal changes of the vertical and horizontal motion and possible basal uplift and bed
separation (e.g. Sugiyama et al, 2004; Nienow et al., 2005). Here, we assume that these changes do not
influence the annual velocities. In our study, the point surface mass balances and vertical velocities have
been measured at the end of the ablation season. As a consequence, the geodetic annual surface mass
balances obtained from the vertical velocities should not be affected by seasonal changes.

562       **6.3 Uncertainties on geodetic point surface mass balances**


The uncertainty related to the point surface mass balance determination results from the uncertainties
on the elevation measurements and on the vertical velocity. Using Equation 4 and assuming
independence of the different sources of uncertainties, the overall uncertainty related to the reconstructed
point surface mass balance is obtained by applying the method of error propagation and assuming
uncorrelated errors:

$\sigma_b{}^2 = 2\sigma_z{}^2 + \sigma_w{}^2$          (6)

in which $\sigma_b$, $\sigma_z$, $\sigma_w$ are the uncertainties relative to the point surface mass balance, elevation and vertical
velocity, respectively.





The uncertainty in elevation depends both on the method of XY positioning, the surface slope or
roughness and the method of altitude determination. Depending on the surface roughness, we can assess
the elevations with an accuracy ranging from 0.1 to 0.3 m from UAV measurements as shown in this
study.
The uncertainty in vertical velocity is $\pm$ 0.1 m a$^{-1}$, as mentioned in the Data section. However, additional
uncertainty could come from the method of elevation observations for the bottom of the stakes. Indeed,
the GNSS measurements are commonly related to the surface of the ice at the location of the stakes and
not to the summit of the stakes. Consequently, the altitude of the bottom  of each stake results from the
difference between the altitude of the surface and the buried height of the ablation stake. Indeed, this
determination is accurate only if the measurement of emergence has been performed exactly from the
point on which the GNSS measurement was made. Unfortunately, in most cases, one operator held the
stick of the GPS antenna at the ice surface close to the ablation stake and another operator measured the
emergence of the stake, but not exactly from the surface altitude that corresponds to the bottom tip of
the GPS antenna. Except for the measurements performed at 2,350 m a.s.l between 2016 and 2019,
which were designed for this purpose, this gives an additional uncertainty of $\pm$ 0.1 m for the altitude of
the bottom of the stake, *i.e.* $\pm 0.14$ m a$^{-1}$ for the calculated vertical velocity.
The overall uncertainty in the geodetic point surface mass balance obtained from remote sensing data is
therefore estimated to range between $\pm 0.20$ and $\pm 0.60$ m a$^{-1}$ using accurate DEMs from UAV
photogrammetry.

**7   Conclusions**
The classical way to determine the point surface mass balance in the ablation zone of a glacier is to set
up ablation stakes and dig pits or conduct drillings in the accumulation zone.
Here, we showed that, in the ablation zone, the point surface mass balances can be reconstructed from
surface altitudes and horizontal velocities only, provided that the vertical velocities have been measured
for at least one year in the past. Our method first requires accurate measurement of the vertical velocities
between two years t and t+1 from stakes and GNSS measurements. Then, for the following or previous
years, the point surface mass balances can be obtained easily from surface measurements only, using
the horizontal velocity and the surface elevation at each end of the horizontal displacement vector (Eq.
4). These measurements can be obtained from remote sensing provided that the ice flow velocity and
altitude determinations are sufficiently accurate.
Our method assumes that the annual vertical velocities are almost constant with time. We have used a
numerical modelling study to show that this approximation holds in areas of low bedrock slope
(averaged averaged over a distance greater than the ice thickness ). This is supported by our detailed
observations performed on Argentière Glacier at 2,400 m a.s.l. and designed for this purpose.
Comparison between the reconstructed point surface mass balances and the observed values shows close



agreement. From our results, we conclude that the point surface mass balances can be obtained with an
accuracy of about 0.3 m w.e. a$^{-1}$ using remote sensing measurements and assuming that the vertical
velocities have been observed accurately over the previous years. Note that the measurement uncertainty
related to the *in situ* measurements of point surface mass balance is 0.14 m w.e. a$^{-1}$ in the ablation zone
(Thibert et al., 2008).
Further tests performed on dataset in other regions of the Argentière and Mer de Glace glaciers show
standard deviations of ±0.2 to ±0.4 m w.e. a$^{-1}$ between reconstructed and observed point surface mass
balances, despite the fact that these measurements were not designed for this purpose. For these tests,
we used the averaged vertical velocities obtained over the last decade.
Given the recent improvements in satellite sensors, it is conceivable to apply our method using high
spatial resolution satellite images like Pléiades or WorldView (0.5 m resolution). For these point surface
mass balance reconstructions, note that, given the strong spatial variability of vertical velocity, it is
crucial to determine the altitudes of the surface at each end of the horizontal displacement vector at the
exact sites on which the vertical velocities are known. We conclude that our method could be useful to
determine numerous point surface mass balances and reduce the amount of effort required to conduct
field measurements, especially in remote areas.
Previous studies have shown that the point surface mass balance signal reveals a climatic signal that is
unbiased by the dynamic glacier response, unlike the commonly used glacier-wide mass balance
(Rasmussen, 2004; Huss et al., 2009; Eckert et al., 2011; Thibert et al., 2018; Vincent et al., 2017). In
the glaciological community, there is growing awareness that point surface mass balance measurements
are important basic data to be shared for mass balance and climate change analyses. In this respect, the
World Glacier Monitoring Service has started collecting such data on a systematic basis as a complement
to glacier-wide surface mass balances [WGMS, 2015]. Our method should open up new prospects to
obtain more numerous point surface mass balances in the future while reducing the amount of time and
energy required for *in situ* measurements.
Another line of research, not explored in the present study, could also be examined. The method
proposed in the present study requires the vertical velocity to reconstruct the annual point surface mass
balance. However, if we derive Equation 4 and assume that vertical velocity is constant with time, we
can determine the surface mass balance changes, instead of the absolute surface mass balances, with the
elevation determinations only. Assuming that satellite sensors provide sufficient accuracy in elevation
and horizontal velocity, this method could be very helpful to reconstruct changes in surface mass balance
in remote areas for which *in situ* measurements are very difficult. In this way, point surface mass balance
changes on numerous unobserved glaciers could be considered with remote sensing observations only.
This would make it possible to obtain climatic signals all over the world, unbiased by dynamic glacier
response.



**Data availability:** The surface mass balance, ice flow velocities and DEM data can be accessed upon request by contacting Christian Vincent (christian.vincent@univ-grenoble-alpes.fr).

**Author contributions:** DC, OL, DS, BJ, LA, UN, AW, LP, OG, VP, ET, FB and CV performed the topographic measurements (photogrammetry, lidar, GNSS). OL, DC, AG, FG, OG, FB and CV performed the numerical calculations and the analysis. AG performed the numerical modelling calculations. CV supervised the study and wrote the paper. All co-authors contributed to discussion of the results.

**Competing interests**: the authors declare that they have no conflict of interest.

**Acknowledgments**

This study was funded by *Observatoire des Sciences de l'Univers de Grenoble* (OSUG) and *Institut des Sciences de l'Univers* (INSU-CNRS) in the framework of the French GLACIOCLIM (*Les GLACIers, un Observatoire du CLIMat*) program. This study was also funded by *l'Agence Nationale de la Recherche* in the framework of the SAUSSURE program (Sliding of glAciers and sUbglacial water pressure (https://saussure.osug.fr) (ANR 18 CE1 0015 01). IGE and ETGR are part of LabEx OSUG@2020 (Investissements d'avenir – ANR10 LABX56). We thank all those who conducted the field measurements. We are grateful to H. Harder for reviewing the English.

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
