# Peer review of "Geodetic point surface mass balances: A new approach to determine point surface mass balances from remote sensing measurements"

_The Cryosphere, 2020_

## Referee Comment (RC1) · Anonymous Referee #1 · 24 Oct 2020

Review of „Geodetic point surface mass balances: A new approach to determine point surface mass balances from remote sensing measurements" by C. Vincent et al.

This manuscript presents a method to derive glacier point surface mass balances from vertical velocities and surface elevation changes. In contrast to similar techniques using the emergence velocity, this method avoids the problem of determining the representative surface slope. In this respect, this new approach circumvents a considerable error source, because surface roughness and medium scale undulations obstruct the effective glacier surface slope. However, the problem remains to accurately determine the vertical velocity at the glacier surface, while the horizontal surface velocity can be more easily derived from remote sensing information. The presented method has a high potential for enabling large scale surface mass balance surveys and the manuscript clearly presents this potential also with respect to the usage of remote sensing information. However, the main difficulty of the validity of the vertical velocity in space and time is not fully investigated so far. It should be possible to use sensitivity experiments based on interpolated parameter fields, to demonstrate the potential errors, which are unavoidably introduced by relying on spatially and temporally discrete remote sensing data. This would allow to evaluate the feasibility of this method in a much better way. In the following, some improvement are suggested, which very likely are rather easily implemented.

Structure: The numerical analysis should be included main analysis, not in the discussion, as this an important component of the overall concept.

Title: It is not obvious that the paper deals with glacier mass balance, even though it is submitted to a cryosphere related journal.

Given the multitude of available data, I am missing a more rigorous analysis of the possibilities by using the parameter fields. All of the parameters show homogeneous spatial fields, even though the local gradients might be large. Therefore, missing point data could also be derived from the spatially interpolated data, which might enable a larger flexibility. This is also true for the temporal evolution. It is stated that the method works with observations of the vertical velocity during periods previous of the elevation and velocity change determination. But there is no analysis, how the temporal change in vertical velocity (and there is a non-negligible trend observed) might impact on the results. The numerical analysis might provide very valuable insight how temporal trends could even be anticipated for certain geometric conditions.

Abstract:

L. 57-64: It is correct that point mass balance measurements represent the balance between local accumulation and ablation and thus a resultant climate information. However, this part of the manuscript provides a rather simplistic description of the situation. Point mass balances need to be measured at constant locations, not along moving stakes for multiple years. What about long term accumulation observations at stable locations? Do we need many distributed measurements across a glacier for resolving climatic information? I know that some of the questions are partly answered in the earlier publications of C. Vincent, but a short summarizing discussion might be helpful.

L. 66-68: There is an objective and an aim of the manuscript. These two sentences point in the same direction and could be combined.

L. 77: The area determination is 17 years old. Is there a newer area estimate?

L. 79: The tributaries are facing SW.

L. 88: It is more likely 2350-2400 m?

L. 97: What does "accurately" mean here: accurately at the end of the ablations season, or accurately like "highly precise"?

L. 100ff: Even though the strategy is explained later, it should be made clear here that the stakes are re-drilled each year at the original position in order to maintain the local reference system. What happens to the remaining stakes (10 m will not melt out at each location every year)? Are they also measured in the following year?

L. 108: 0.01 m accuracy is rather optimistic, given the short occupation times (what about resolving multipath uncertainties from 60 observations?).

L. 110 ff: This paragraph starts with vertical velocities, without introducing the requirements of using velocities at all. Maybe it is better to insert a sentence that both velocity components are required. The "bottom tip" of the stake means the "real bottom" where the lowermost stake segment touches the ice? "Emergence measurements" mean ablation measurements? How can you be sure about the tilt of the stakes in the borehole? Does this tilt change over time? How do you obtain the same level of accuracy for horizontal and vertical velocities, while the z-component of GNSS measurements are usually not as precise as the horizontal ones?

L. 120: This is probably "focal length", not "focal lens".

L. 123: These are probably "resulting" and not "original" ground resolutions. The original ground resolution of the photographs might be higher (smaller dimension).

L. 130: Accuracy information for the location and elevation of the ortho-mosaics and the DEMs are missing.

L. 134: I am confused. In line 123, the dimension of the ortho-mosaic is given with 0.1 m. There is no need for resampling then.

L. 141: But this also depends on the quality of the ortho-images and their co-registration, which is not provided.

Fig. 1: It would be instructive, to have isohypses across the glacier, in order to see the exact location, as the numbers only indicate a broad region. Also for Mer de Glace, some isohypses would be helpful.

Fig. 2: The colour coding should also be included as legend in the figure itself.

L. 159: space between "framework" and "used".

L. 170: the surface mass balance needs to be expressed in the same dimension/material as the other components.

L. 173: the downslope direction is a bit misleading, as local slope patterns might show different directions as the main flow. The statement is only true for the mean slope over a certain distance.

L. 179/180: Well, slope can be calculated along any distance. But this is a critical point of the entire theory: which is the appropriate scale of surface slope for such analysis. I am not sure that the annual displacement is the correct scale. This requires some elaboration.

L. 186: This conclusion tells us that the determination of the emergency velocities has rather large errors.

L. 203: It should be noted that all vectors in this diagram have the unit of velocity: m/yr.

L. 217ff: It took me quite a while to digest this statement. Finally, I think the strong point of this formulation is that measurements are taken at the annual displacement distance. In consequence, the relative thickness change is based on identical geometric and surface conditions. A small scale surface undulation is detected at exactly the same relative location and therefore does not influence the elevation change. Also surface conditions, like patches of lower albedo, are advected and do not alter the ablation conditions. Maybe this should be elaborated.

L. 229-240: The description of velocity measurements and interpolation of the velocity field is not fully clear. First, it seems to me that a larger number of stakes were not drilled at the last-years location in 2018 (stakes 12 and 14-19). Even if this is mentioned later, it should also noted here, because these are not negligible deviations. As the surface velocity field of a glacier is rather homogeneous (which is also documented in Fig. 5), the measurement location has no large influence on the interpolated velocity field, as long as the measurement density is sufficient. However, the exact location of the stake is important for the application of the presented theory.

L. 243: Which two periods do you refer to?

Fig. 5b: In my opinion, the larger differences of the stakes with offsets in the relocation are only due to larger uncertainties in the velocity determination. In principle, temporal deviations in the vertical velocity field (not in the point measurements) should be expressed in an analog manner as in the horizontal velocity field due to the incompressibility condition of ice.

L. 281ff: Here you use the slope along the 1-year displacement vector, correct? This is probably not an appropriate choice, even for a smooth glacier section.

L. 313: You should provide a reasoning, why you use the vertical velocities of the previous year, instead for the year of the mass balance measurements.

L. 317-324: This observation reflects the situation that the vertical velocity field shows considerable spatial gradients. It would be interesting to see how the results change if you use the values from the interpolated field at the exact measurement locations.

L. 328: This 15 m is probably site related and should be discussed.

Fig. 9: As far as I can see, the vertical velocities are measured at the midpoints of the annual displacement vectors. This is different from the method described in Fig. 3, where the vertical velocity is determined for the downstream displacement vector. How does this influence the resuts?

Fig. 10: The isohypses are very thin and hard to see. I am not sure what additional information is provided by this figure. It is also not referenced in the text.

L. 404f: I do not understand this remark, as it is stated in the introduction that a noticeable debris cover is only observed below the ice fall.

L. 407f: Does this infer that the vertical velocity is determined for each single year from GNSS measurements and then the mean value for 2001-2018 is used, based on the fact that the stakes were replaced regularly within a distance of 35m?

L. 461f: This argument is not correct, as can be seen in Fig. 11b. But the changes are rather smooth and comparably small, but definitely not negligible.

L. 480: Again, small is a rather relative condition. Chages from 0.2 to -0.5 m/yr within one year (Fig. 11b, stake 2) are hardly small.

L. 483 onward: In my opinion, this section belongs to methods and results, respectively, as this is an essential part of the paper and should not be presented in the discussion.

---

## Referee Comment (RC2) · Ben Pelto (Referee) · 30 Oct 2020

**Review of "Geodetic point surface mass balances: A new approach to determine point surface mass balances from remote sensing measurements" by C. Vincent et al.**

Vincent et al. present a method to derive glacier point surface mass balances from vertical ice velocities and surface elevation changes. Their method avoids the large uncertainties associated with determining representative surface slope with which to calculate emergence velocities. Typically, surface roughness and irregular larger-scale glacier surface topography account for considerable uncertainty in slope estimates. By eliminating this large error source, this method reduces uncertainty on estimates of geodetic point surface mass balance. Determining vertical velocity at the glacier surface remains a challenge, which here, the authors measure at ablation stakes. Their method demonstrates the potential for expanding the limited number of point observations available globally of surface mass balance, which are labor-intensive. The authors demonstrate that their method can also be used with remote sensing information—necessary for wider applicability of this approach. The challenge of well-representing the vertical velocity, particularly with respect to time, requires further attention. If attended to, this method represents a valuable contribution to the glaciological community.

There are numerous uses for this method beyond the primary aim, including the establishment of new records of mass balance, or the filling of data gaps in glaciological records. When new glaciological records are established, this method could be applied to extend the point mass balance record to the years preceding the in situ record by collecting geodetic data until in situ measurements can begin. Further, glaciological observations for some glaciers, or some portions of some glaciers, are incomplete in some years, due to logistical or other challenges. This year (Covid-19) offers one such example for some glacier records. This method would allow for point mass balance to be determined from only remote sensing information for given points or a given glacier, avoiding the issue of gaps in valuable long-term records.

Like Reviewer 1, I agree that some form of a sensitivity analysis regarding the spatial and temporal representation of vertical ice velocities would be beneficial, and not onerous to conduct. I elaborate this point in comments below.

I also find it interesting that the trend of vertical ice velocity decrease seems relatively constant e.g. Figure 11, and that the potential bias introduced by assuming constant vertical ice velocity may in part be accounted for by applying a empirically-based decrease-rate factor (perhaps via horizontal velocities using the ratio of horizontal ice velocity to vertical ice velocity for a given area (either modeled or observed)) to represent the decline in vertical ice velocities expected to accompany horizontal ice velocity over decadal-scales.

Below I present specific line-comments.

**Specific comments**

1 Add "glacier" to the title.

L 54-60 Are valid statements, though it should be highlighted that a series of point surface mass balance observations, e.g. across an entire glacier or elevation band, can be considered a direct climate signal. Individual point balances may indeed respond to climate,

but may represent local processes (wind scour, avalanching, etc.). Perhaps this should be briefly discussed.

L107-109 Were there any observations taken to constrain this error? It is often useful to test a few control points with the same method (occupation length etc.) to assess uncertainty.

L112 Emergence measurements seems confusing to me. This refers to stake height, or stake protrusion, correct? I would re-word for clarity, as emergence velocity is used throughout this manuscript, it is confusing to use emergence to describe measuring a different quantity, even though the word is correctly used here.

L133-135 Resampled from 1.0 m to 0.1 m? But I thought the ortho was 0.1 m-resolution and then used to produce a 1.0 m-resolution DEM. Perhaps clarify.

L149 The contours are nearly invisible. Either make them stand out more or reduce their number (larger interval). The blue and green dots are difficult to make out as well.

L174 Perhaps down-glacier direction instead of downslope direction, local slopes will often be upslope but down-glacier.

L217-219 Yes, and perhaps most importantly, will not be affected by the advection of surface topography, that is, if we measured a given point through the year, crevasses, surface roughness, supraglacial streams, etc, may be advected over a given point, but your formulation, measuring a stake embedded in the ice, avoids these complications.

L264 Nice graphic, it seems to me that the vertical ice velocity is in fact changing over the three-year period, with a decrease across the three years, as can be seen in the horizontal velocities in the figure as noted in L242-244. The vertical velocities are decreasing with the horizontal velocity decrease.

L281 It may be valuable to describe how slope was determined, between the two GPS survey locations? From the DEM? From slope measurements around the two survey points? It may be worthwhile to test using different methods to determine slope, if remote methods can be used, does this represent the slope better, or not? Either way the conclusion will be of value.

L298 Figure 7. Certainly greater dispersion, but the comparison does not look unfavorable. The decrease in emergence velocity through time can be seen with the red dots below the black. Why not add in the regression lines?

L337 Figure axes labels are difficult to read at this size. Perhaps use only a single y-axis label and slightly increase font size for all text.

L377 remove extra period

L517-530 This section describes the competing factors which influence vertical velocities well. Overall, the authors make a compelling argument for minor changes in vertical ice velocity. However, two primary issues arise from their formulation: 1) that this method is only suitable for relatively low-angle glacier terrain, which implies that this method can primarily only be applied for valley glacier tongues; and 2) that while the change in vertical ice velocity is

indeed minor, that it may not be negligible. As the authors point out, the horizontal ice velocity decreased by around 4% per year---regardless of whether this trend were to continue---such a rate of decrease over a decade is substantial, and thus is cannot be assumed that vertical ice velocity is stable over decadal scales. Decreasing ice velocity has been observed for many glaciers around the globe (Dehecq et al., 2019; Heid and Kääb, 2012), and given the current rate of ice wastage, that is, disequilibrium of glaciers (Christian et al., 2018; Zemp et al., 2015), assuming stable vertical ice velocities is questionable. Figure 11 highlights this, with vertical velocity falling by 0.5 m a$^{-1}$ to 1.0 m a$^{-1}$ over a decade which likely would present a non-negligible bias in assessing surface mass balance from remote data with this method over decadal periods.  As the authors state, part of the decrease in vertical ice velocity will be compensated by reduced ice flux convergence/divergence produced by bedrock topography.

L469 An uncertainty of 0.2 m w.e. a$^{-1}$ seems optimistic for decadal periods, but accurate for short periods, like the three-year window of this study. Perhaps it would be best to state this directly, that surface mass balance can be obtained from this method with an accuracy of about 0.2 m w.e. a$^{-1}$ over periods of 1-5? years, but over periods of 5-10+ years with an accuracy of XX m w.e. With the XX value determined by calculating the uncertainty or bias in using one year's vertical ice velocity to calculate mass balance for years in the 5-10 year range for stakes where that length of record is available in this study.

L591 It is not clear what the range represents: 0.2 m w.e. if the elevation accuracy is determined to be 0.1 m and 0.6 m w.e. if it is determined to be 0.3 m? This is a critical point that should be expanded upon. If this method is to be applied elsewhere—e.g. with other remote datasets, what accuracy/resolution is needed, or how will uncertainty scale with reduced accuracy/resolution?

L616 Change "dataset" to "datasets".

Citations: I suggest adding DOIs to all references for which one exists. Currently only some entries have a listed DOI, and some DOIs are "https:…" and others just the DOI itself. Ensure consistency with the TC formatting guidelines.

**References**

Christian, J. E., Koutnik, M. and Roe, G. H.: Committed retreat: controls on glacier disequilibrium in a warming climate, Journal of Glaciology, 64(246), 675–688, doi:10.1017/jog.2018.57, 2018.

Dehecq, A., Gourmelen, N., Gardner, A. S., Brun, F., Goldberg, D., Nienow, P. W., Berthier, E., Vincent, C., Wagnon, P. and Trouvé, E.: Twenty-first century glacier slowdown driven by mass loss in High Mountain Asia, Nature Geoscience, 12(1), 22–27, doi:10.1038/s41561-018-0271-9, 2019.

Heid, T. and Kääb, A.: Repeat optical satellite images reveal widespread and long term decrease in land-terminating glacier speeds, The Cryosphere, 6(2), 467–478, doi:10.5194/tc-6-467-2012, 2012.

Zemp, M., Frey, H., Gärtner-Roer, I., Nussbaumer, S. U., Hoelzle, M., Paul, F., Haeberli, W., Denzinger, F., Ahlstrøm, A. P., Anderson, B., Bajracharya, S., Baroni, C., Braun, L. N.,

Cáceres, B. E., Casassa, G., Cobos, G., Dávila, L. R., Delgado Granados, H., Demuth, M. N., Espizua, L., Fischer, A., Fujita, K., Gadek, B., Ghazanfar, A., Hagen, J. O., Holmlund, P., Karimi, N., Li, Z., Pelto, M., Pitte, P., Popovnin, V. V., Portocarrero, C. A., Prinz, R., Sangewar, C. V., Severskiy, I., Sigurðsson, O., Soruco, A., Usubaliev, R. and Vincent, C.: Historically unprecedented global glacier decline in the early 21st century, Journal of Glaciology, 61(228), 745–762, doi:10.3189/2015JoG15J017, 2015.

---

## Author Comment (AC1) · 9 Dec 2020

Response to Reviewers:

We thank the Reviewers for their comments and suggestions to improve this manuscript. We address their comments below. Reviewers comments are in italics, and our responses are in normal font below. Changes to the text have been highlighted in the revised manuscript.

Response to Reviewer 1

General comments

a) Review of An Geodetic point surface mass balances: A new approach to determine point surface mass balances from remote sensing measurements Az by C. Vincent et al. This manuscript presents a method to derive glacier point surface mass balances from vertical velocities and surface elevation changes. In contrast to similar techniques using the emergence velocity, this method avoids the problem of determining the representative surface slope. In this respect, this new approach circumvents a considerable error source, because surface roughness and medium scale undulations obstruct the effective glacier surface slope. However, the problem remains to accurately determine the vertical velocity at the glacier surface, while the horizontal surface velocity can be more easily derived from remote sensing information. The presented method has a high potential for enabling large scale surface mass balance surveys and the manuscript clearly presents this potential also with respect to the usage of remote sensing information. However, the main difficulty of the validity of the vertical velocity in space and time is not fully investigated so far. It should be possible to use sensitivity experiments based on interpolated parameter fields, to demonstrate the potential errors, which are unavoidably introduced by relying on spatially and temporally discrete remote sensing data. This would allow to evaluate the feasibility of this method in a much better way. In the following, some improvement are suggested, which very likely are rather easily implemented.

Thanks for these comments. We agree that the vertical velocity and its change with time and space is a crucial point. We provide a response to these comments below (see in particular answer to comment (d)).

b) Structure: The numerical analysis should be included main analysis, not in the discussion, as this an important component of the overall concept.

Writing this paper, we hesitated to include the numerical analysis in the Results Section. Finally, we decided not to include it because, as shown in Figure 13a, the vertical velocities are not well reproduced (Fig. 13a and inset of Fig. 14a) using the Elmer/Ice model, although the pattern of horizontal velocities is well reproduced (Fig.
13b). Consequently, our numerical experiments were mainly used to analyse the temporal changes in vertical velocities and to understand why the observed pattern of surface vertical velocity is quite steady over time. The numerical analysis is therefore helpful to understand the causes of the temporal changes in vertical velocities but not reliable enough to accurately reconstruct the spatial pattern of the vertical velocities. We believe that the main conclusions of this paper come from observations and that the numerical analysis is helpful for their interpretation only. It explains why the numerical analysis is not included in the main analysis.

c) Title: It is not obvious that the paper deals with glacier mass balance, even though it is submitted to a cryosphere related journal.

Agree. The title has been modified by adding "on glaciers"

d) Given the multitude of available data, I am missing a more rigorous analysis of the possibilities by using the parameter fields. All of the parameters show homogeneous spatial fields, even though the local gradients might be large. Therefore, missing point data could also be derived from the spatially interpolated data, which might enable a larger flexibility. This is also true for the temporal evolution. It is stated that the method works with observations of the vertical velocity during periods previous of the elevation and velocity change determination. But there is no analysis, how the temporal change in vertical velocity (and there is a non-negligible trend observed) might impact on the results. The numerical analysis might provide very valuable insight how temporal trends could even be anticipated for certain geometric conditions.

It is a crucial point indeed. The uncertainties relative to the spatial and temporal changes of vertical velocities are discussed in different sections in the manuscript and we acknowledge that it may lead to confusion. In addition, we acknowledge that the temporal trend is not analysed accurately from our observations and not discussed rigorously enough. We suggest to complete this analysis according to the following analysis:

TCD
Regarding the spatial variations: Our detailed observations from the stake network used between 2016 and 2018 at Argentière glacier (2350 m a.s.l.) showed that the vertical velocity change can exceed 0.3 m a-1 if the stakes are located at distances of more than 25 or 30 meters (section 5.1). This conclusion come from the errors relative to the locations of the stakes (some stakes are located at distances of more than 25 meters from the initial positions). In section 5.3, we showed that the surface mass balance can be reconstructed with an accuracy of about 0.2 m w.e. a-1 using the vertical velocities observed within a radius of less than 15 m. The whole network suggest that the vertical velocity spatial gradient can exceed 1.5 m a-1/100 m in this region. As a consequence, a horizontal deviation of 10 m could lead to a vertical velocity change exceeding the measurement uncertainty (0.15 m a-1). It seems not reasonable to interpolate the vertical velocity from measurements performed 100 m away from each other. For the new version of the manuscript, additional observations have been analyzed (new Figure S1) in order to better assess the vertical velocity spatial gradient over length scales of 20 to100 m. For this purpose, the vertical velocities have been calculated from 10 stakes set up in 2018/2019 on a longitudinal profile located between the stakes 3 and 13 (see Figure 2 for the locations of these stakes). Note that the distances between these stakes are small and enable to assess the vertical velocity variations at small scale. According to these measurements shown in the following Figure, the spatial gradient can reach 0.02 a-1. It is a little larger than what we found previously (0.015 a-1). However, it does no change the main conclusion: in order to reconstruct the surface mass balance from remote sensing, it requires measurement of the horizontal ice flow velocity and the altitudes of the ends of the velocity vector exactly at the same location, within a radius of less than 15 m compared to that of vertical velocity determination. However, further detailed and numerous observations would be needed to better assess the spatial gradient of the vertical velocities at the scale of 10 - 20 m. Finally, we can conclude that, although the general spatial changes of the vertical velocity shown in Figure S1 seem homogeneous, a detailed examination shows that the vertical velocity cannot be interpolated with an accuracy better than 0.3

**TCD**
or 0.4 m a-1 from measurements performed 100 m away from each other.

Caption of Figure S1 (included in the Supplementary of the new version of this paper): Vertical velocities measured from 10 stakes set up in 2018/2019 on a longitudinal profile located between the stakes 3 and 13 (see Figure 2 for the locations of the stakes 3 and 13).

Regarding the temporal changes :

It is not easy to analyse accurately the temporal changes of the vertical velocities from our observations given that (i) our detailed observations performed at Argentière glacier (2350 m) between 2016 and 2018 is not long enough to study the temporal changes. Note however that the temporal changes over the 3 years observations does not reveal temporal changes exceeding the measurements uncertainties as shown in Figure 5b and explained in Section 5.1, (ii) the longer series of observations available to study the temporal changes were not designed to measure the vertical velocities. For this reason, the following conclusions should be regarded with some caution until better data becomes availabe. From the longer series of observations performed at Argentière glacier at 2550 m and 2700 m a.s.l. (Fig. 11b), we assessed a general temporal trend of about 0.07 m a-2. We can conclude that the past period on which we have determined the vertical velocities should no exceed 4 years in order to not exceed an uncertainty of 0.3 m w.e. a-1 on the reconstructed surface mass balance. This conclusion could be different with stronger temporal change in vertical velocities. Another idea could be to assess the temporal change in vertical velocities from the temporal change in horizontal velocities and to apply the same ratio. Unfortunately, the changes in vertical and horizontal velocity observed at Argentière glacier at 2550 m and 2700 m a.s.l. (Fig. 11b) are very different, 2-3% a-1 and 1.5 % a-1 respectively. Further observations and analyses are needed to clarify this point.

To reply to this comment, we completed this analysis and summarized the impact of spatial and temporal changes in vertical velocities on the reconstructed surface mass
balance uncertainties in Section 6.2. In addition, we added some sentences in the Conclusion to summarize the main conclusions of this analysis. In Section 6.2: "Our dataset shows that vertical velocities strongly vary in space over the glacier surface. Our detailed observations from the network used between 2016 and 2018 at the Argentière Glacier (2350 m) showed that the vertical velocity change can exceed 0.3 m a-1 if the stakes are located at distances of more than 25 or 30 meters (section 5.1). We showed that the surface mass balance can be reconstructed with an accuracy of about 0.2 m w.e. a-1 using the vertical velocities observed within a radius of less than 15 m. Records from the whole network suggest that the vertical velocity spatial gradient can exceed 1.5 m a-1/100 m in this region. As a consequence, a horizontal deviation of 10 m could lead to a vertical velocity change exceeding the measurement uncertainty (0.15 m a-1). To better assess the vertical velocity spatial gradient over length scales of 20 to100 m, the vertical velocities have been calculated from 10 stakes set up in 2018/2019 on a longitudinal profile located between stakes 3 and 13 (Fig. 2). Note that the distances between these stakes is small and enable to assess the vertical velocity variations at small scales. According to measurements shown in Figure S1, the spatial gradient can reach up to 0.02 a-1, which is slightly more important than what we found previously (0.015 a-1). We can conclude that reconstructing surface mass balance from remote sensing requires measurements of the horizontal ice flow velocity and the altitudes of the ends of the velocity vector exactly at the same locations, i.e. within a radius of less than 15 m compared to that of vertical velocity determination. The analysis of temporal changes also deserves particular attention. The 3 years of detailed observations performed at 2350 m at Argentière Glacier does not reveal temporal changes exceeding the measurement uncertainties, as shown in Figure 5b. Note that the longer series of observations available to study the temporal changes over decadal time scales were not designed to measure the vertical velocities. However, from the longer series of observations performed at Argentière glacier at 2550 m and 2700 m a.s.l. (Fig. 11b), we assessed a general temporal trend of about 0.07 m a-2. We can conclude that the past period over which the vertical velocities are determined
should not exceed 4 years in order to not exceed an uncertainty of 0.3 m w.e. a-1 on the reconstructed surface mass balance. This conclusion could be different with stronger temporal change in vertical velocities. Further observations and analysis are needed to better estimate the temporal changes." In Conclusion, we added: "From our results, we conclude that the point surface mass balances can be obtained with an accuracy of about 0.3 m w.e. a-1 using remote sensing measurements and assuming that the vertical velocities have been observed accurately over the previous years within a radius of less than 15 m. We also conclude, from our datasets that the past period over which the vertical velocities are determined should not exceed 4 years in order to not exceed an uncertainty of 0.3 m w.e. a-1 for the reconstructed surface mass balance, although further observations and analysis are needed to better estimate these spatial and temporal changes."

Introduction:

L. 57-64: It is correct that point mass balance measurements represent the balance between local accumulation and ablation and thus a resultant climate information. However, this part of the manuscript provides a rather simplistic description of the situation. Point mass balances need to be measured at constant locations, not along moving stakes for multiple years. What about long term accumulation observations at stable locations? Do we need many distributed measurements across a glacier for resolving climatic information? I know that some of the questions are partly answered in the earlier publications of C. Vincent, but a short summarizing discussion might be helpful.

Ok. We added some explanations in the introduction : "Ablation is related directly to the surface energy balance. Accumulation is related to solid precipitation but is also strongly influenced by valley topography. Indeed, glaciers are generally surrounded by very steep non-glacial slopes which capture precipitation over a larger area than that of the glacier itself. In this way, high accumulation values are due to downhill transportation and strong winds actions [e.g. Vincent, 2002]. Statistical modelling enables us to extract a climatic signal from a heterogeneous in-situ observations of

TCD
point mass balance networks independently of effects related to ice flow dynamics and glacier area changes (Vincent et al., 2018b). However, these previous studies showed that it is crucial to perform observations of point annual surface mass balance at the same locations every year."

L. 66-68: There is an objective and an aim of the manuscript. These two sentences point in the same direction and could be combined.

Agree. It has been reformulated : Âń In this way, we aim at determining point surface mass balances in ablation areas without setting up ablation stakes each year."

L. 77: The area determination is 17 years old. Is there a newer area estimate?

Agree. The surface area was assessed at 10.9  $\rm km^2$  in 2018. It has been changed in the new version.

L. 79: The tributaries are facing SW.

Agree. It has been changed.

L. 88: It is more likely 2350-2400 m?

Agree. It has been changed.

L. 97: What does "accurately" mean here: accurately at the end of the ablations season, or accurately like "highly precise"?

Right. We suggest : " with a high positioning accuracy" to clarify this sentence.

L. 100ff: Even though the strategy is explained later, it should be made clear here that the stakes are re-drilled each year at the original position in order to maintain the local reference system. What happens to the remaining stakes (10 m will not melt out at each location every year)? Are they also measured in the following year?

Agree. We added a sentence in the manuscript : Âń We performed the observations of point annual surface mass balance at the same locations each year Âż. The obser-
vations of the remaining stakes are not used in this study because they do not allow to cover the whole year until the end of the ablation season.

L. 108: 0.01 m accuracy is rather optimistic, given the short occupation times (what about resolving multipath uncertainties from 60 observations?).

We performed tests from several measurements on the same fixed point during the day. If the antenna is fixed on a base which is attached on a rock outside the glacier (i.e without movement of antenna and base), the accuracy is better than 0.01 m provided that the number of visible satellites is greater than 7 and the distance between fixed and mobile receivers is less than 1 km. This is the intrinsic accuracy (the manufacturers usually guarantee better accuracy). It does not take into account the possible tilt of the stick supporting the antenna and others factors which could affect the accuracy of the measurements. Concerning our observations, the main source of uncertainty is not the intrinsic of accuracy of the GNSS instruments but it is related to the size of the boreholes and the possible tilt of the stakes

L. 110 ff: This paragraph starts with vertical velocities, without introducing the requirements of using velocities at all. Maybe it is better to insert a sentence that both velocity components are required. The "bottom tip" of the stake means the "real bottom" where the lowermost stake segment touches the ice? "Emergence measurements" mean ablation measurements? How can you be sure about the tilt of the stakes in the borehole? Does this tilt change over time? How do you obtain the same level of accuracy for horizontal and vertical velocities, while the z-component of GNSS measurements are usually not as precise as the horizontal ones ?

Some changes have been done in this paragraph in order to improve the explanations : "Both velocity components are required. The vertical velocity is the vertical component of the surface velocity obtained from measuring altitude differences of the bottom tip of stakes. For this purpose, the emergence measurement is required to obtain the buried length of the stake. Thus, the purpose of emergence observations is two-fold. They
enable (i) to calculate the surface mass balance from two field campaigns and, (ii) to obtain the altitude of the bottom tip of the stake using the altitude of the surface. In practice, the DGPS measurements are performed simultaneously with the emergence measurements in order to obtain the exact position of the bottom tip of the stake buried in ice. In this way, it is possible to monitor ice velocity along the three directions. Depending on the tilt of the ablation stakes in the borehole, the size of the drilling hole and the mechanical play of the jointed stakes, we assume that the annual horizontal and vertical velocities are known with an uncertainty of  $\pm$  0.10 m a-1." To the question : Ân Does the tilt of the stakes in the borehole change over time ? Âz, we can reply that the deformation close to the surface (depth less than 10 m) can be neglected over period of one year. About the last question related to the accuracy of vertical and horizontal coordinates, we note that we estimate the same level of accuracy for horizontal and vertical velocities, because the main uncertainty does not depend on the intrinsic accuracy of DGPS instruments. It depends mainly on the initial tilt of the ablation stakes, the size of the drilling hole and the mechanical play of the jointed stakes.

L. 120: This is probably "focal length", not "focal lens".

The change has been done.

L. 123: These are probably "resulting" and not "original" ground resolutions. The original ground resolution of the photographs might be higher (smaller dimension).

Agree. It has been changed.

L. 130: Accuracy information for the location and elevation of the ortho-mosaics and the DEMs are missing.

Agree. Some information have been added.

L. 134: I am confused. In line 123, the dimension of the ortho-mosaic is given with 0.1 m. There is no need for resampling then.
Agree, it was confusing. In the new version, we wrote: "The horizontal resolutions of the ortho-photo mosaics and digital elevation models (DEMs) are 10 cm and 1.0 m, respectively." And latter "). Due to the velocities of the Argentière glacier in this region ( $\sim$ 55 m a-1), we resampled the UAV ortho-photo at 1.0 m resolution

L. 141: But this also depends on the quality of the ortho-images and their co-registration, which is not provided.

The surveys were acquired using 10 common GCPs located on off-glacier stable area, no coregistration step was used. In addition, we measured the uncertainities over 25 random points given results of  $\pm 0.55$ m over these stable areas. Figure 2 show the borders off-glacier areas where the points were measured. The quality of orthomosaics are not so different, except for the presence of shadows that can affect the correlation. Nonetheless, the fact of resampling orthomosaics, allow to reduce this effect. Further information has been added in the manuscript.

Fig. 1: It would be instructive, to have isohypses across the glacier, in order to see the exact location, as the numbers only indicate a broad region. Also for Mer de Glace, some isohypses would be helpful.

We did not add contour lines in Figure 1 for the sake of clarity. However, we added the elevations of each zone, including the Tacul glacier in the Mer de Glace basin.

Fig. 2: The colour coding should also be included as legend in the figure itself.

Agree. It has been done.

L. 159: space between "framework" and "used".

Done

L. 170: the surface mass balance needs to be expressed in the same dimension/material as the other components.

Yes, it is the case. bs the surface mass balance is expressed in meters of ice, firn or
snow (m a-1). S the surface elevation is expressed in meters of ice, firn or snow (m). us, vs, ws the components of ice flow velocity at the surface are expressed in meters of ice, firn or snow (m a-1). In our case, bs ,  $\partial S/\partial t$ , us, vs, ws are expressed in meters of ice per year given that we use the annual values at the end of the ablation season (no firn, no snow). L. 173: the downslope direction is a bit misleading, as local slope patterns might show different directions as the main flow. The statement is only true for the mean slope over a certain distance.

Agree. Here, we replaced Âń downslope direction Âż by Âń flow direction Âż. And we added a sentence stating that we assume the downslope direction being the flow direction.

L. 179/180: Well, slope can be calculated along any distance. But this is a critical point of the entire theory: which is the appropriate scale of surface slope for such analysis. I am not sure that the annual displacement is the correct scale. This requires some elaboration.

Unlike other glaciological problems in which the slope can be selected for different distances, the distance on which the slope is calculated for the emergence velocities should correspond to the requirements of Equation 3. It is crucial to calculate the slope for a given year, from elevations measurements at the two GNSS survey locations, whatever the method used (remote sensing or in-situ observations) in order to respect Equation 3. If we use the slope of the year t, i.e tan $\alpha$ t , we have to use  $\Delta$ h2, which is the annual thickness change observed at the end of the annual ice flow vector. Conversely, if we use the slope of the year t+1, i.e tan $\alpha$ t+1, we have to use  $\Delta$ h1, which is the annual thickness change observed at the beginning of the annual ice flow vector (Fig. 3). Thus, we do not think that there is an "appropriate scale" of surface slope for such analysis.

However, the large uncertainties related to the slope and thickness changes prevent us from calculating the point surface mass balance from the emergence velocities, as
explained some lines later.

L. 186: This conclusion tells us that the determination of the emergency velocities has rather large errors.

Yes. Morever, it is discussed in detail in Section 6.1. At the end of this sentence, we added the reference to Section 6.1 to be clearer.

L. 203: It should be noted that all vectors in this diagram have the unit of velocity: m/yr.

Ok. The units have been added.

L. 217ff: It took me quite a while to digest this statement. Finally, I think the strong point of this formulation is that measurements are taken at the annual displacement distance. In consequence, the relative thickness change is based on identical geometric and surface conditions. A small scale surface undulation is detected at exactly the same relative location and therefore does not influence the elevation change. Also surface conditions, like patches of lower albedo, are advected and do not alter the ablation conditions. Maybe this should be elaborated.

We are not sure to understand the comment relative to Âń the relative thickness change is based on identical geometric and surface conditions. A small scale surface undulation is detected at exactly the same relative location and therefore does not influence the elevation change Âż. As shown in Equation 4, the reconstructed surface mass balance depends on the elevations of the surface at each end of the ice flow vector and on the vertical velocity only. Consequently, it does not depend on the surface slope that can change from one year to the next, or from one site to another, neither on thickness changes that can vary from one site to another. We are afraid that more explanations would be confusing.

L. 229-240: The description of velocity measurements and interpolation of the velocity field is not fully clear. First, it seems to me that a larger number of stakes were not drilled at the last-years location in 2018 (stakes 12 and 14-19). Even if this is mentioned

TCD
later, it should also noted here, because these are not negligible deviations. As the surface velocity field of a glacier is rather homogeneous (which is also documented in Fig. 5), the measurement location has no large influence on the interpolated velocity field, as long as the measurement density is sufficient. However, the exact location of the stake is important for the application of the presented theory.

We agree that the text was not very clear . We are sorry. We changed the text and tried to reformulate the paragraph in order to clarify this point.

L. 243: Which two periods do you refer to?

The two periods 2016/2017- 2017/2018 and 2017/2018-2018/2019. It has be clarified in the new version.

Fig. 5b: In my opinion, the larger differences of the stakes with offsets in the relocation are only due to larger uncertainties in the velocity determination. In principle, temporal deviations in the vertical velocity field (not in the point measurements) should be expressed in an analog manner as in the horizontal velocity field due to the incompressibility condition of ice.

We disagree with this comment. In Data section, we estimated that the annual horizontal and vertical velocities are known with an uncertainty of  $\pm$  0.10 m a-1. The small dots shown Fig. 5b show larger deviations without any bias (positive or negative). In addition, they correspond to the stakes that were set up at distances of more than 25 m from the initial positions. Although the vertical velocity changes could be affected by the horizontal motion changes or vertical strain rate changes as discussed in Section 6, the large differences observed here are very likely related to the positions of the stakes.

L. 281ff: Here you use the slope along the 1-year displacement vector, correct? This is probably not an appropriate choice, even for a smooth glacier section.

The distance on which the slope is calculated for the emergence velocities should cor-

TCD
respond to the requirements of Equation 3. As explained above, it is crucial to calculate the slope for a given year, from elevations measurements at the two GNSS positioning surveys, whatever the method used (remote sensing or in-situ observations) in order to respect Equation 3 (see also the reply to comment I.179/180 above)

L. 313: You should provide a reasoning, why you use the vertical velocities of the previous year, instead for the year of the mass balance measurements.

We use the vertical velocities observed during the previous year in 2016-2017 in order to test the method. The use of vertical velocities observed in 2017-2018, i.e the same year of reconstructed mass balance, would not provide any error given that the emergence measurements used for surface mass balance determination are also used for the vertical velocities. The reconstructed mass balance would be exactly the observed mass balance. In this case, the Equation 3 is perfectly solved. Here, the topic is to assess the uncertainty obtained on the reconstructed mass balance when we use independent data related to vertical velocities coming from previous years. We do not believe that more explanations are required.

L. 317-324: This observation reflects the situation that the vertical velocity field shows considerable spatial gradients. It would be interesting to see how the results change if you use the values from the interpolated field at the exact measurement locations.

The reply to this comment is similar to the reply of comment d) above. Although the general spatial changes of the vertical velocity shown in Figures 4 and S1 seem homogeneous, a detailed examination shows that the vertical velocity cannot be interpolated with an accuracy better than 0.3 or 0.4 m a-1 from measurements performed 100 m away from each other.

L. 328: This 15 m is probably site related and should be discussed.

This question has been discussed in the response of the previous comment. As explained above, we suggest to add this new analysis in the new version in Section 6.2
and we added some sentences in Conclusions.

Fig. 9: As far as I can see, the vertical velocities are measured at the midpoints of the annual displacement vectors. This is different from the method described in Fig. 3, where the vertical velocity is determined for the downstream displacement vector. How does this influence the results?

Yes, the vertical velocities are measured at the midpoints of the annual displacement vectors. Figure 3 shows the same thing : the vertical velocity ws is obtained from the elevations difference of the bottom tip of the stake between the year t and the year t+1. In this way, ws is the average of the vertical velocity we could observe between the point 1 and the point 2. We apply for Fig. 9 the exact same method described in Fig. 3. Another point is the date of measurements. As mentioned at the end of Section 6.2, it is crucial to calculate the annual velocities from measurements performed at the end of the ablation season in order to get free of the seasonal changes of the vertical and horizontal motion. Here, we assume that these changes do not influence the annual velocities because the point surface mass balances and vertical velocities have been measured at the end of the ablation season. As a consequence, the geodetic annual surface mass balances obtained from the vertical velocities should not be affected by seasonal changes.

Fig. 10: The isohypses are very thin and hard to see. I am not sure what additional information is provided by this figure. It is also not referenced in the text.

The contour lines have been changed. The reference to Figure 10 has been added in the Section 5.4 of the manuscript. "Then we used the DEMs from 2018 and 2019 (Fig. 10) to determine the elevations of these points Zs1, 2018 and Zs2, 2019 (see Eq. 4 and Figure 3)."

L. 404f: I do not understand this remark, as it is stated in the introduction that a noticeable debris cover is only observed below the ice fall.
Yes, a noticeable debris cover is observed below the ice fall with debris cover which can reach more than 50 cm. In the studied region at 2350 m, the ice is generally free of debris. Although the debris cover does not exceed 5 to 10 cm, these differences can lead to significant surface roughness. We added an explanation in the Study Area Section. "In the detailed studied region at 2350 m, the ice is generally free of debris. The debris cover can be 5 to 10 cm thick in some locations."

L. 407f: Does this infer that the vertical velocity is determined for each single year from GNSS measurements and then the mean value for 2001-2018 is used, based on the fact that the stakes were replaced regularly within a distance of 35m?

Yes, we used the mean value of the vertical velocity obtained for the period 2001-2018. Yes, the stakes were replaced regularly (but not each year) within a distance of 35m. Additional explanations have been added to clarify this point.

L. 461f: This argument is not correct, as can be seen in Fig. 11b. But the changes are rather smooth and comparably small, but definitely not negligible.

Agree. The changes in vertical velocity are not significant at 2350 m over the period 2016-2019 but it is not true for longer period on the other sites. Over decadal time scale, it seems that the temporal changes are small but not negligible. These sentences have been changed in the manuscript and it is discussed now in the new analysis of Section 6.2 about spatial and temporal changes.

L. 480: Again, small is a rather relative condition. Chages from 0.2 to -0.5 m/yr within one year (Fig. 11b, stake 2) are hardly small.

Agree. As explained above, we added a thorough analysis and summarized the impact of spatial and temporal changes in vertical velocities on the reconstructed surface mass balance uncertainties in Section 6.2.

L. 483 onward: In my opinion, this section belongs to methods and results, respectively, as this is an essential part of the paper and should not be presented in the

TCD
discussion. See response to general comment (b).

Please also note the supplement to this comment: https://tc.copernicus.org/preprints/tc-2020-239/tc-2020-239-AC1-supplement.pdf

**TCD**
the ablation stakes used in this study for annual surface mass balance and ice flow velocity measurement.
**Fig. 2.** Map of the studied area in the ablation zone of Argentière glacier. The contour lines of surface topography correspond to the surface in 2018.
**Fig. 3.** Diagram illustrating horizontal, vertical and emergence velocities (m a-1) observed from an ablation stake (orange).
Fig. 4. Horizontal (top panel) and vertical (bottom) ice flow velocities (m a-1) measured over

three years from the ablation stakes. Note the different colour scales. Distances in m.
**Fig. 5.** Comparison of horizontal ice flow velocities (a) and vertical velocities (b) between the years 2016/2017, 2017/2018 and 2018/2019. The black dots correspond to the comparison between the 2016/2017and
Fig. 6. Emergence velocities between the years 2016/2017, 2017/2018 and 2018/2019 (m a-1)

**Fig. 7.** Comparison of emergence velocities between the years 2016-2017, 2017-2018 and 2018-2019. The black dots correspond to the comparison between the 2016-2017and 2017-2018 periods. The red dots correspond
**Fig. 8.** Observed and calculated point surface mass balances at 2,350 m a.s.l. at Argentière glacier. The point surface mass balances have been calculated: a) for the year 2017-2018 using the vertical velociti
**Fig. 9.** Horizontal velocities obtained from feature tracking (Cosi-Corr) using UAV images. The black crosses show the locations where the vertical velocities were observed. The red dots correspond to the ends

CCC ①

lines). The black dots correspond to the positions of the stakes in 2018 and 2019 observed from

GNSS measurements.
Fig. 11. Horizontal (a) and vertical (b) velocities observed at the different stakes at 2,550 m

a.s.l.(Stakes 2 and 3) and 2,700 m a.s.l. (stakes 7, 8, 9, 10, 11 and 12).

Interactive

comment

**Fig. 12.** Observed and calculated point surface mass balances from: a) two ablation stakes located at 2,550 m a.s.l. at Argentière glacier measured between 2002 and 2018, b) six stakes located at around 2,700 m

TCD

**Fig. 13.** Vertical (a) and horizontal (b) surface velocities modelled at Argentière glacier in 2015. Red dots show the locations of the ablation stakes set up at 2,400 m and 2,650 m a.s.l.

**Fig. 14.** Modelled changes in vertical (a) and horizontal (b) surface velocities between 1998 and 2015. Insets compare modelled velocities at the stake location (orange dots) between 1998 and 2015.

**Fig. 15.** Modelled changes in vertical velocities at the surface (a) and at the bedrock (b) between 1998 and 2015. The righthand figure (c) shows the change in vertical velocity at the surface due to change in

**Supplement:**

---

## Author Comment (AC2) · 9 Dec 2020

Response to Reviewers:

We thank the Reviewers for their comments and suggestions to improve this manuscript. We address their comments below. Reviewers comments are in italics, and our responses are in normal font below. Changes to the text have been highlighted in the revised manuscript.

Response to Reviewer 2

General comments

Vincent et al. present a method to derive glacier point surface mass balances from vertical ice velocities and surface elevation changes. Their method avoids the large uncertainties associated with determining representative surface slope with which to calculate emergence velocities. Typically, surface roughness and irregular larger-scale glacier surface topography account for considerable uncertainty in slope estimates. By eliminating this large error source, this method reduces uncertainty on estimates of geodetic point surface mass balance. Determining vertical velocity at the glacier surface remains a challenge, which here, the authors measure at ablation stakes. Their method demonstrates the potential for expanding the limited number of point observations available globally of surface mass balance, which are labor-intensive. The authors demonstrate that their method can also be used with remote sensing information—necessary for wider applicability of this approach. The challenge of well-representing the vertical velocity, particularly with respect to time, requires further attention. If attended to, this method represents a valuable contribution to the glaciological community. There are numerous uses for this method beyond the primary aim, including the establishment of new records of mass balance, or the filling of data gaps in glaciological records. When new glaciological records are established, this method could be applied to extend the point mass balance record to the years preceding the in situ record by collecting geodetic data until in situ measurements can begin. Further, glaciological observations for some glaciers, or some portions of some glaciers, are incomplete in some years, due to logistical or other challenges. This year (Covid-19) offers one such example for some glacier records. This method would allow for point mass balance to be determined from only remote sensing information for given points or a given glacier, avoiding the issue of gaps in valuable long-term records. Like Reviewer 1, I agree that some form of a sensitivity analysis regarding the spatial and temporal representation of vertical ice velocities would be beneficial, and not onerous to conduct. I elaborate this point in comments below. I also find it interesting that the trend of vertical ice velocity decrease seems relatively constant e.g. Figure 11, and that the potential bias introduced by assuming constant vertical ice velocity may

in part be accounted for by applying a empirically-based decrease-rate factor (perhaps via horizontal velocities using the ratio of horizontal ice velocity to vertical ice velocity for a given area (either modeled or observed)) to represent the decline in vertical ice velocities expected to accompany horizontal ice velocity over decadal-scales.

The reply to this comment is similar to the reply we have done to Reviewer 1. It is a crucial point indeed.

The uncertainties relative to the spatial and temporal changes of vertical velocities are discussed in different sections in the manuscript and we acknowledge that it may lead to confusion. In addition, we acknowledge that the temporal trend is not analysed accurately from our observations and not discussed rigorously enough. We suggest to complete this analysis according to the following analysis :

Regarding the spatial variations : Our detailed observations from the network used between 2016 and 2018 at Argentière glacier (2350 m) showed that the vertical velocity change can exceed 0.3 m a-1 if the stakes are located at distances of more than 25 or 30 meters (section 5.1). This conclusion come from the errors relative to the locations of the stakes (some stakes are located at distances of more than 25 meters from the initial positions). In section 5.3, we showed that the surface mass balance can be reconstructed with an accuracy of about 0.2 m w.e. a-1 using the vertical velocities observed within a radius of less than 15 m. The whole network suggest that the vertical velocity spatial gradient can exceed 1.5 m a-1/100 m in this region. As a consequence, a horizontal deviation of 10 m could lead to a vertical velocity change exceeding the measurement uncertainty (0.15 m a-1). It seems not reasonable to interpolate the vertical velocity from measurements performed 100 m away from each other. For the new version of the manuscript, additional observations have been analyzed (new Figure S1) in order to better assess the vertical velocity spatial gradient over length scales of 20 to100 m. For this purpose, the vertical velocities have been calculated from 10 stakes set up in 2018/2019 on a longitudinal profile located between the stakes 3 and 13 (see Figure 2 for the locations of these stakes). Note that the distances between

these stakes are short and enable to assess the vertical velocity variations at small scale. According to these measurements shown in the following Figure, the spatial gradient can reach 0.02 a-1. It is a little more important than that we found previously (0.015 a-1). However, it does no change the main conclusion: in order to reconstruct the surface mass balance from remote sensing, it requires measurement of the horizontal ice flow velocity and the altitudes of the ends of the velocity vector exactly at the same location, within a radius of less than 15 m compared to that of vertical velocity determination. However, further detailed and numerous observations would be needed to better assess the spatial gradient of the vertical velocities at the scale of 10 − 20 m. Although the general spatial changes of the vertical velocity shown in Figure S1 seem homogeneous, a detailed examination shows that the vertical velocity cannot be interpolated with an accuracy better than 0.3 or 0.4 m a-1 from measurements performed 100 m away from each other.

Caption of Figure S1 (included in the Supplementary of the new version of this paper ): Vertical velocities measured from 10 stakes set up in 2018/2019 on a longitudinal profile located between the stakes 3 and 13 (see Figure 2 for the locations of the stakes 3 and 13).

Regarding the temporal changes :

It is not easy to accurately analyse the temporal changes of the vertical velocities from our observations given that (i) our detailed observations performed at Argentière glacier (2350 m) between 2016 and 2018 is not long enough to study the temporal changes. Note however that the temporal changes over the 3 years observations does not reveal temporal changes exceeding the measurements uncertainties as shown in Figure 5b and explained in Section 5.1, (ii) the longer series of observations available to study the temporal changes were not designed to measure the vertical velocities. For this reason, the following conclusions should be regarded with some caution until better data becomes availabe. From the longer series of observations performed at Argentière glacier at 2550 m and 2700 m a.s.l. (Fig. 11b), we assessed a general temporal

trend of about 0.07 m a-2. We can conclude that the past period on which we have determined the vertical velocities should no exceed 4 years in order to not exceed an uncertainty of 0.3 m w.e. a-1 on the reconstructed surface mass balance. This conclusion could be different with stronger temporal change in vertical velocities. Another idea could be to assess the temporal change in vertical velocities from the temporal change in horizontal velocities and to apply the same ratio. Unfortunately, the changes in vertical and horizontal velocity observed at Argentière glacier at 2550 m and 2700 m a.s.l. (Fig. 11b) are very different, 2-3% a-1 and 1.5 % a-1 respectively. Further observations and analysis are needed to clarify this point.

To reply to this comment, we completed this analysis and summarized the impact of spatial and temporal changes in vertical velocities on the reconstructed surface mass balance uncertainties in Section 6.2. In addition, we added some sentences in the Conclusion to summarize the main conclusions of this analysis.

Specific comments

1 Add "glacier" to the title.

It has been done.

L 54-60 Are valid statements, though it should be highlighted that a series of point surface mass balance observations, e.g. across an entire glacier or elevation band, can be considered a direct climate signal. Individual point balances may indeed respond to climate, but may represent local processes (wind scour, avalanching, etc.). Perhaps this should be briefly discussed.

Agree. Some explanations have been added in the new version of the manuscript

L107-109 Were there any observations taken to constrain this error? It is often useful to test a few control points with the same method (occupation length etc.) to assess uncertainty.

We performed tests from several measurements on the same fixed point during the

day. If the antenna is fixed on a base which is attached on a rock outside the glacier, the accuracy is better than 0.01 m provided that the number of visible satellites is greater than 7 and the distance between fixed and mobile receivers is less than 1 km. This is the intrinsic accuracy (the manufacturers usually guarantee better accuracy). It does not take into account the possible tilt of the antenna and others factors which could affect the accuracy of the measurements. Concerning our observations, the main source of uncertainty is not the intrinsic accuracy of the GNSSS instruments but is related to the size of the boreholes and the possible tilt of the stakes

L112 Emergence measurements seems confusing to me. This refers to stake height, or stake protrusion, correct? I would re-word for clarity, as emergence velocity is used throughout this manuscript, it is confusing to use emergence to describe measuring a different quantity, even though the word is correctly used here.

The emergence measurement refers to the stake protrusion. The emergence observations enable (i) to calculate the surface mass balance from two field campaigns and, (ii) to obtain the altitude of the bottom tip of the stake using the altitude of the surface. We tried to clarify this point in the new version.

L133-135 Resampled from 1.0 m to 0.1 m? But I thought the ortho was 0.1 m-resolution and then used to produce a 1.0 m-resolution DEM. Perhaps clarify.

Agree. It was an error. The initial resolution of ortho-mosaic was 0.1m and we resampled it to 1.0 m. It has been changed.

L149 The contours are nearly invisible. Either make them stand out more or reduce their number (larger interval). The blue and green dots are difficult to make out as well.

Figure 2 has been improved.

L174 Perhaps down-glacier direction instead of downslope direction, local slopes will often be upslope but down-glacier.

Agree. We provide more explanation to clarify this point. Here, before the Equation 2,

we wrote Âń If the horizontal x-axis is taken in the flow direction. . ..". and four lines later, we wrote "In this way we assume that the downslope direction is the flow direction. Âż.

L217-219 Yes, and perhaps most importantly, will not be affected by the advection of surface topography, that is, if we measured a given point through the year, crevasses, surface roughness, supraglacial streams, etc, may be advected over a given point, but your formulation, measuring a stake embedded in the ice, avoids these complications.

Ok

L264 Nice graphic, it seems to me that the vertical ice velocity is in fact changing over the three-year period, with a decrease across the three years, as can be seen in the horizontal velocities in the figure as noted in L242-244. The vertical velocities are decreasing with the horizontal velocity decrease.

Yes, but note that the vertical velocity can be positive or negative as seen in Figure 4b or Figure 5b. Consequently, a decrease in horizontal velocity leads to an increase in vertical velocity (if the vertical velocity is negative) or a decrease in vertical velocity (if the vertical velocity is positive). The absolute value of vertical velocity should decrease but the consequence on the reconstructed mass balance (Equation 4) is not always in the same way. Some further explanations are provided in Section 6.2.

L281 It may be valuable to describe how slope was determined, between the two GPS survey locations? From the DEM? From slope measurements around the two survey points? It may be worthwhile to test using different methods to determine slope, if remote methods can be used, does this represent the slope better, or not? Either way the conclusion will be of value.

Between the two GNSS positioning surveys, the slope was determined from the Digital Elevation Model using UAV measurements, between the two GPS survey locations. It has been clarified in the new version. It is crucial to calculate the slope for a given year, from elevations measurements at the two GNSS survey locations, whatever the

method used (remote sensing or in-situ observations) in order to resoect Equation 3. If we use the slope of the year t, i.e $\tan \alpha t$ , we have to use $\Delta h2$, which is the annual thickness change observed at the end of the annual ice flow vector. Conversely, If we use the slope of the year t+1, i.e $\tan \alpha t+1$ , we have to use $\Delta h1$, which is the annual thickness change observed at the beginning of the annual ice flow vector.

L298 Figure 7. Certainly greater dispersion, but the comparison does not look unfavorable. The decrease in emergence velocity through time can be seen with the red dots below the black. Why not add in the regression lines?

Agree. Unfortunately, the regression lines do not allow pointing out a significant change between the first year and the following years. We did not add the regression lines in ordre not to overload the Figure.

L337 Figure axes labels are difficult to read at this size. Perhaps use only a single y-axis label and slightly increase font size for all text.

It has been done

L377 remove extra period

Agree

L517-530 This section describes the competing factors which influence vertical velocities well. Overall, the authors make a compelling argument for minor changes in vertical ice velocity. However, two primary issues arise from their formulation: 1) that this method is only suitable for relatively low-angle glacier terrain, which implies that this method can primarily only be applied for valley glacier tongues; and 2) that while the change in vertical ice velocity is indeed minor, that it may not be negligible. As the authors point out, the horizontal ice velocity decreased by around 4% per year— regardless of whether this trend were to continue—such a rate of decrease over a decade is substantial, and thus is cannot be assumed that vertical ice velocity is stable over decadal scales. Decreasing ice velocity has been observed for many glaciers

around the globe (Dehecq et al., 2019; Heid and Kääb, 2012), and given the current rate of ice wastage, that is, disequilibrium of glaciers (Christian et al., 2018; Zemp et al., 2015), assuming stable vertical ice velocities is questionable. Figure 11 highlights this, with vertical velocity falling by 0.5 m a-1 to 1.0 m a-1 over a decade which likely would present a non-negligible bias in assessing surface mass balance from remote data with this method over decadal periods. As the authors state, part of the decrease in vertical ice velocity will be compensated by reduced ice flux convergence/divergence produced by bedrock topography.

To reply to this comment and to the general comments, we completed this analysis and summarized the impact of spatial and temporal changes in vertical velocities on the reconstructed surface mass balance uncertainties in Section 6.2. See our detailed reply to the general comments.

L469 An uncertainty of 0.2 m w.e. a-1 seems optimistic for decadal periods, but accurate for short periods, like the three-year window of this study. Perhaps it would be best to state this directly, that surface mass balance can be obtained from this method with an accuracy of about 0.2 m w.e. a-1 over periods of 1-5? years, but over periods of 5-10+ years with an accuracy of XX m w.e. With the XX value determined by calculating the uncertainty or bias in using one year's vertical ice velocity to calculate mass balance for years in the 5-10 year range for stakes where that length of record is available in this study.

Agree. It has been clarified. Moreover, the final uncertainties are mentioned in Conclusions

L591 It is not clear what the range represents: 0.2 m w.e. if the elevation accuracy is determined to be 0.1 m and 0.6 m w.e. if it is determined to be 0.3 m? This is a critical point that should be expanded upon. If this method is to be applied elsewhere—e.g. with other remote datasets, what accuracy/resolution is needed, or how will uncertainty scale with reduced accuracy/resolution?

Agree. It has been clarified. In addition, the requirements and the final uncertainties are mentioned in Conclusions.

L616 Change "dataset" to "datasets".

It has been done

Citations: I suggest adding DOIs to all references for which one exists. Currently only some entries have a listed DOI, and some DOIs are "https:..." and others just the DOI itself. Ensure consistency with the TC formatting guidelines.

It has been done

References Christian, J. E., Koutnik, M. and Roe, G. H.: Committed retreat: controls on glacier disequilibrium in a warming climate, Journal of Glaciology, 64(246), 675–688, doi:10.1017/jog.2018.57, 2018. Dehecq, A., Gourmelen, N., Gardner, A. S., Brun, F., Goldberg, D., Nienow, P. W., Berthier, E., Vincent, C., Wagnon, P. and Trouvé, E.: Twenty-first century glacier slowdown driven by mass loss in High Mountain Asia, Nature Geoscience, 12(1), 22–27, doi:10.1038/s41561-018-0271-9, 2019. Heid, T. and Kääb, A.: Repeat optical satellite images reveal widespread and long term decrease in land-terminating glacier speeds, The Cryosphere, 6(2), 467–478, doi:10.5194/tc-6-467-2012, 2012. Zemp, M., Frey, H., Gärtner-Roer, I., Nussbaumer, S. U., Hoelzle, M., Paul, F., Haeberli, W., Denzinger, F., Ahlstrøm, A. P., Anderson, B., Bajracharya, S., Baroni, C., Braun, L. N., Cáceres, B. E., Casassa, G., Cobos, G., Dávila, L. R., Delgado Granados, H., Demuth, M. N., Espizua, L., Fischer, A., Fujita, K., Gadek, B., Ghazanfar, A., Hagen, J. O., Holmlund, P., Karimi, N., Li, Z., Pelto, M., Pitte, P., Popovnin, V. V., Portocarrero, C. A., Prinz, R., Sangewar, C. V., Severskiy, I., SigurÃřsson, O., Soruco, A., Usubaliev, R. and Vincent, C.: Historically unprecedented global glacier decline in the early 21st century, Journal of Glaciology, 61(228), 745–762, doi:10.3189/2015JoG15J017, 2015.

Please also note the supplement to this comment:

https://tc.copernicus.org/preprints/tc-2020-239/tc-2020-239-AC2-supplement.pdf

[Figure]

**Supplement:**

---

## Author Response (AR2)

**Christian VINCENT**
Institut des Géosciences de l'Environnement,
BP 96 38402 St Martin d'Hères Cedex
Tél : (33) 4 76 82 42 47
Email : christian.vincent@univ-grenoble-alpes.fr

To

Chief Editor of « The Cryosphere »

Grenoble, 22 january 2021

Dear Chief Editor,

Please, find enclosed the new version of the manuscript "Geodetic point surface mass balances: A new approach to determine point surface mass balances from remote sensing measurements" from *Christian Vincent, Diego Cusicanqui, Bruno Jourdain, Olivier Laarman, Delphine Six, Adrien Gilbert, Andrea Walpersdorf, Antoine Rabatel, Luc Piard, Florent Gimbert, Olivier Gagliardini, Vincent Peyaud, Laurent Arnaud, Emmanuel Thibert, Fanny Brun and Ugo Nanni,* that we submit to the Cryosphere.

The required corrections (Figure 14 and "Reviewer" ) have been done.
Thanks again for your work, your comments and suggestions.

Yours sincerely,

Christian Vincent